# AI-Researcher: Autonomous Scientific Innovation

**Jiabin Tang*   Lianghao Xia*   Zhonghang Li   Chao Huang†**
The University of Hong Kong
{jiabintang77, bjdwh.zzh, chaohuang75}@gmail.com; aka_xia@foxmail.com

## Abstract

The powerful reasoning capabilities of Large Language Models (LLMs) in mathematics and coding, combined with their ability to automate complex tasks through agentic frameworks, present unprecedented opportunities for accelerating scientific innovation. In this paper, we introduce AI-Researcher, a fully autonomous research system that transforms how AI-driven scientific discovery is conducted and evaluated. Our framework seamlessly orchestrates the complete research pipeline–from literature review and hypothesis generation to algorithm implementation and publication-ready manuscript preparation–with minimal human intervention. To rigorously assess autonomous research capabilities, we develop Scientist-Bench, a comprehensive benchmark comprising state-of-the-art papers across diverse AI research domains, featuring both guided innovation and open-ended exploration tasks. Through extensive experiments, we demonstrate that AI-Researcher achieves remarkable implementation success rates and produces research papers that approach human-level quality. This work establishes new foundations for autonomous scientific innovation that can complement human researchers by systematically exploring solution spaces beyond cognitive limitations. Code link: https://github.com/HKUDS/AI-Researcher.

## 1 Introduction

Scientific discovery has historically been constrained by human cognitive limitations and the immense scale of potential solution spaces Wang et al. [2023]. Recent advances in Large Language Models (LLMs) have demonstrated remarkable capabilities in mathematical reasoning, coding, and problem-solving tasks that were previously thought to require human expertise Didolkar et al. [2024], Guo et al. [2024]. However, transitioning from isolated capabilities to fully autonomous scientific research systems capable of original innovation remains an unsolved challenge that could fundamentally transform how scientific progress occurs.

Despite recent advances in agentic frameworks powered by LLMs, scientific innovation represents an intellectual frontier orders of magnitude more challenging than the task automation currently mastered by existing AI agents Manus Technologies [2025], OpenManus Contributors [2025], Li et al. [2023], Tang et al. [2025]. While today's agents can schedule meetings or retrieve structured information, genuine scientific discovery demands an unprecedented level of intelligence—requiring sophisticated conceptual reasoning across abstract theoretical domains, transformative hypothesis generation that bridges disparate knowledge fields, and methodological innovation that extends far beyond pattern recognition. The research process necessitates maintaining coherent understanding across thousands of papers while simultaneously generating insights that fundamentally advance knowledge boundaries—intellectual capabilities that existing architectures cannot approach.

Most critically, scientific exploration involves navigating vast, unbounded solution spaces with deeply uncertain rewards, requiring meta-cognitive abilities to recognize promising directions and abandon

---

*Equal contribution.

†Chao Huang is the Corresponding Author.

39th Conference on Neural Information Processing Systems (NeurIPS 2025).

unproductive paths. Researchers must continuously evaluate experimental results against theoretical frameworks, adapt hypotheses based on unexpected findings, and communicate complex ideas with precision and clarity—all while maintaining the creative spark that drives breakthrough discoveries. These profound limitations have prevented AI systems from autonomously conducting meaningful scientific work, perpetuating a paradigm where AI remains relegated to narrow assistance roles rather than functioning as independent scientific contributors capable of accelerating human knowledge advancement through systematic exploration of solution spaces beyond human cognitive limitations.

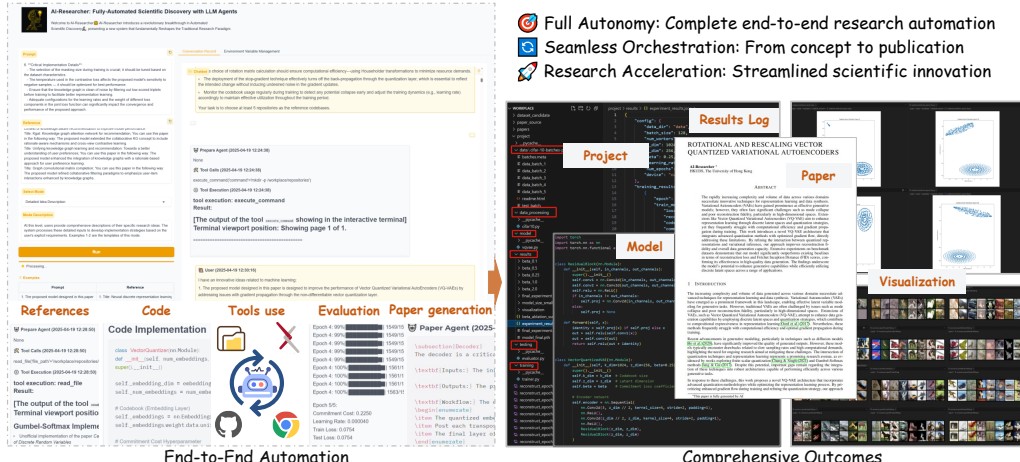

Figure 1: Architectural overview of AI-Researcher, illustrating the end-to-end autonomous scientific innovation pipeline encompassing literature exploration, idea generation, algorithm implementation, experimental validation, and comprehensive scholarly publication with rigorous evaluation metrics.

While specialized systems exist for literature analysis or experiment design Schmidgall and Moor [2025], Gottweis et al. [2025], they fail to orchestrate the complete research workflow from hypothesis generation through publication-quality reporting. Furthermore, no standardized benchmarks exist to evaluate autonomous research across diverse scientific domains, making progress in this frontier difficult to measure systematically.

We introduce AI-Researcher a novel framework that addresses these limitations by seamlessly orchestrating the complete scientific discovery lifecycle—from literature analysis through implementation to scholarly documentation. Unlike systems focusing on isolated capabilities, our framework employs a comprehensive multi-agent architecture where specialized components collaborate through structured knowledge exchange to maintain coherent reasoning throughout the research process. This recursive refinement mechanism enables continuous bidirectional feedback between theoretical concepts and their implementations—preserving intellectual consistency while transforming research ideas into rigorous scientific contributions with minimal human intervention.

AI-Researcher introduces three key innovations that fundamentally advance autonomous scientific discovery. **First**, Resource Analyst agents decompose complex research concepts into atomic components with explicit bidirectional mappings between mathematical formulations and code implementations, dramatically reducing hallucination risks. **Second**, our Implementation Framework employs a human-inspired iterative refinement paradigm where specialized agents collaborate through structured feedback cycles, mirroring the proven mentor-student relationship in academic research. **Third**, our Documentation Agent overcomes LLM coherence limitations through a hierarchical synthesis approach that transforms research artifacts into publication-quality manuscripts while maintaining cross-document consistency and factual integrity throughout extensive scholarly documentation.

To rigorously evaluate autonomous scientific systems, we develop **Scientist-Bench**—the first comprehensive benchmark enabling standardized assessment across both guided innovation scenarios and open-ended exploration tasks spanning diverse AI domains. Through extensive experiments on 22 benchmark papers using multiple LLM evaluators, we demonstrate that AI-Researcher achieves remarkable implementation success rates while producing research contributions that frequently approach human-level quality. Surprisingly, our findings reveal AI-Researcher performs better in open-ended exploration than in guided implementation tasks—suggesting autonomous research systems excel when leveraging internal knowledge synthesis rather than following prescriptive directives.

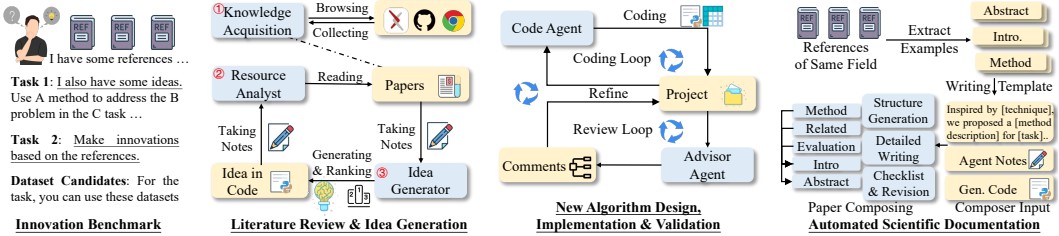

Figure 2: Architectural framework of AI-Researcher.

These results establish new foundations for autonomous scientific agents that complement human researchers by systematically exploring solution spaces beyond human cognitive limitations.

## 2 Scientist-Bench: Benchmarking AI Agents for Scientific Discovery

Scientific discovery requires deep expertise and methodical reasoning. Developing benchmarks for novel scientific discovery and establishing evaluation metrics remain challenges in the field Reddy and Shojaee [2025]. We introduce Scientist-Bench, a comprehensive benchmark comparing LLM Agent-generated research with human expert work. Unlike existing benchmarks Wang et al. [2024], Scientist-Bench provides a comprehensive framework including curated research instructions, references, and datasets derived from peer-reviewed papers. This enables direct comparison between AI-generated and human scientific contributions through multidimensional evaluation standards. We define the scientific discovery task as follows (full benchmark details are in Appendix A.7):

**Task Formulation:** The benchmark evaluates agent systems for scientific research capabilities. The input $\mathcal{X} = \{\mathcal{R}, I, \mathcal{D}\}$ consists of reference papers $\mathcal{R}$ (15-20 relevant references selected via LLMs), research instruction $I$ (containing core research ideas), and datasets $\mathcal{D}$. Scientist-Bench defines two challenge levels: **Level-1** provides explicit research instructions extracted from target paper $y$, testing execution ability; **Level-2** omits these instructions, challenging agents to formulate novel directions using only references and datasets. The output $\hat{\mathcal{Y}} = \{\mathcal{C}, p\}$ comprises implementation code $\mathcal{C}$ and a technical report $p$ describing research background, methodology, experiments, and results. Both components undergo evaluation to measure quality and innovation compared to human-generated research, providing holistic assessment across theoretical and practical dimensions.

## 3 The AI-Researcher Framework

### 3.1 Multi-Agent System Overview of AI-Researcher

Recent work has shown AI systems' potential for autonomous scientific discovery Lu et al. [2024], Yamada et al. [2025]. AI-Researcher builds on this by introducing a systematic framework with three stages: **i) Literature Review and Idea Generation**; **ii) New Algorithm Design, Implementation and Validation**; and **iii) Automated Scientific Documentation**. As shown in Figure 2, this pipeline transforms scientific concepts into complete academic contributions with minimal human oversight.

#### 3.1.1 Literature Review

• **Knowledge Acquisition Agent**. The autonomous research process begins with literature exploration by the `Knowledge Acquisition Agent`, which systematically gathers *Relevant Papers* and *Code Repositories* from scientific databases. A key advantage is its minimal input requirement—users need only provide 10-15 reference papers. The system then processes this input to identify valuable information, performing two critical functions below. These filtering criteria ensure only relevant, maintained, and impactful resources form the foundation for subsequent AI research. Detailed prompts and tools for the `Knowledge Acquisition Agent` are provided in Appendix A.7.

1) **Code Repository Selection**: Using reference papers as guidance, the agent identifies at least 5 high-quality GitHub repositories through filtering that evaluates: Code Recency, GitHub Popularity, Documentation Quality, Domain Relevance, Citation Impact.

2) **Supplementary Literature Gathering**: For each filtered high-quality repository, the agent automatically retrieves corresponding papers from arXiv, including their complete LaTeX source files, further enriching the knowledge base with contextually relevant technical materials.

• **Resource Analyst Agent**. This agent systematically deconstructs research concepts into manageable atomic components, meticulously extracting their mathematical formulations and corresponding code implementations through its **Paper Analyst** and **Code Analyst** sub-agents, ensuring precise alignment between theoretical expressions and practical implementation.

**Secure Research Environment.** To protect host systems during automated operations, all processes run in a Docker container, providing: (1) robust security boundaries preventing unauthorized modifications; (2) consistent environments with pre-configured ML frameworks; and (3) dynamic package management for autonomous dependency installation. This creates a controlled yet flexible workspace supporting the entire research pipeline.

**Integrated Research Analysis**. `Resource Analyst` forms a critical bridge between abstract concepts and their concrete implementations, significantly reducing potential hallucinations in subsequent development stages. This agent operates through a carefully structured analytical process:

1) **Concept Decomposition**: Using the initial research idea, the agent decomposes complex objectives into atomic academic concepts—fundamental, indivisible research elements requiring investigation.

2) **Mathematical Formalization**: The `Paper Analyst` examines LaTeX files through RAG-based paradigm to extract mathematical formulations of each atomic concept.

3) **Implementation Analysis**: The `Code Analyst` analyzes code repositories to locate implementations of these mathematical expressions, identifying critical files and dependencies.

4) **Knowledge Integration**: Results from paper and implementation analyses are synthesized into concept profiles, establishing connections between math formulations and code implementations.

This rigorous cycle continues until all concepts are thoroughly investigated, culminating in a detailed research report that serves as the foundation for development planning. The `Plan Agent` transforms these findings into a comprehensive implementation roadmap addressing training procedures, testing methodologies, and dataset requirements–creating a complete, executable research strategy.

### 3.1.2  Idea Generation

Recent LLMs have advanced research ideation, with Chain-of-Ideas Li et al. [2024] organizing literature into progressive chains and ResearchAgent Baek et al. [2025] using collaborative LLM reviewers to refine proposals. While these systems primarily recombine known knowledge, our `Idea Generator` is designed to venture beyond established paradigms into new scientific frontiers.

Operating after comprehensive analysis, the `Idea Generator` employs knowledge synthesis to identify unexplored research territories. The agent systematically seeks conceptual gaps, contradictory findings, and emerging patterns across literature and implementations—areas where scientific discoveries often emerge. Each generated proposal pushes beyond established paradigms through:

• *Challenges* that pinpoint fundamental limitations in current scientific understanding; • *Existing Methods* revealing conceptual blind spots ripe for innovation; • *Motivation* establishing scientific necessity for paradigm-shifting approaches; • *Proposed Method* introducing novel theoretical frameworks or algorithmic innovations; • *Technical Details* translating abstract breakthroughs into implementable science; and • *Expected Outcomes* projecting potential scientific and practical impact.

**Divergent-Convergent Discovery**. Inspired by Si et al. [2024], our process first generates five distinct research directions in a divergent phase, exploring orthogonal perspectives. These undergo convergent evaluation against *Scientific Novelty*, *Technical Soundness*, and *Transformative Potential*. The top concept is then developed into a comprehensive proposal with clear implementation pathways.

### 3.2  New Algorithm Design, Implementation and Validation

Translating novel research concepts into functioning implementations represents one of the most challenging aspects of computational science. Unlike traditional code agents that attempt one-shot implementations–often causing errors or research misalignment–we introduce a framework that mirrors the proven human research paradigm of iterative refinement and collaborative feedback.

• **Multi-Stage Refinement Architecture.** Our approach implements a cyclical development process with explicit feedback mechanisms, enabling progressive improvements. Similar to advisor-student collaborations, our framework conducts iterative refinement with structured guidance. This approach increases implementation success rates with test-time scaling capabilities.

• **Code Implementation Framework.** The `Code Agent` transforms research analysis and development plans into executable implementations. Operating within a controlled workspace, this agent creates structured implementations with comprehensive file system and execution capabilities. It enforces strict code independence principles while ensuring faithful translation of academic concepts into working code. Throughout development, the agent maintains continuous verification against the implementation plan with thorough documentation of all modifications.

• **Expert Validation Framework.** Our `Advisor Agent` provides expert feedback that bridges the gap between theoretical concepts and practical implementation. It validates implementation fidelity by systematically comparing code against atomic research ideas extracted during analysis. The agent examines results through specialized navigation tools and visualizations while referencing workspace materials. Based on comprehensive evaluation, it generates detailed assessment reports with specific, actionable modification recommendations to guide refinement iterations.

• **Progressive Experimental Cycles.** Our experimental process implements a rigorous scientific approach to code validation. The `Code Agent` begins by developing prototype implementations that undergo initial testing on minimal data (typically 1-2 epochs or small dataset subsets) to establish baseline feasibility. Following this preliminary validation, successful implementations that incorporate review feedback advance to full-scale experiments, while persistently unsuccessful implementations receive "unfeasible" classification after multiple refinement attempts. Throughout this cyclical process, the `Advisor Agent` provides analytical support by evaluating results and recommending supplementary investigations. These recommendations span implementation refinements, validation studies, visualizations, and comparative analyses aligned with established research standards. Through these structured refinement cycles, implementations systematically evolve toward optimal performance, ensuring scientific rigor and reproducibility in our findings.

### 3.3 Automated Scientific Documentation

The culmination of scientific research requires transforming raw experimental results into structured academic knowledge contributions. Following substantial implementation and experimentation cycles, our `Documentation Agent` initiates a sophisticated process that converts technical artifacts into publication-ready manuscripts while maintaining scientific integrity and narrative coherence.

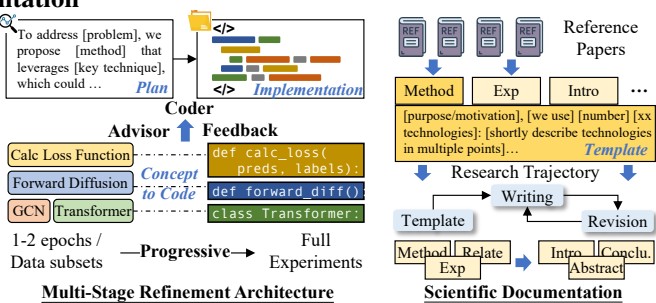

Figure 3: Illustration of (1) multi-stage implementation refinement, and (2) automated sceintfic documentation.

**Research Trajectory Synthesis.** The `Automated Documentation Agent` systematically integrates diverse research elements—including agent reasoning processes, execution logs, implemented code, and experimental outcomes—into cohesive scientific narratives. This holistic approach preserves the complete intellectual context behind discoveries while structuring findings according to established academic conventions. Unlike simple documentation tools, our system captures both the final results and the critical decision pathways that led to scientific advances.

**Overcoming Document-Scale Coherence Challenge.** To write coherent academic manuscripts–a challenge for LLMs that struggle with consistency over extended outputs–we developed a multi-stage generation framework inspired by how researchers draft papers and by Shao et al. [2024]. This methodology overcomes LLM limitations by decomposing writing into manageable components while preserving logical connections and factual integrity throughout.

**Three-Phase Hierarchical Documentation.** Our writing approach employs a systematic three-stage process: (1) **Synthesizing Research Artifacts**: structural outlining based on domain-appropriate templates that establish section hierarchies and logical flow; (2) **Template-Guided Structure**: methodical content elaboration that develops explanations maintaining cross-document consistency; and (3) **Hierarchical Documentation Process**: systematic verification using specialized academic checklists that identify and remediate inaccuracies or omissions. This "one more step" review process enhances factual integrity and completeness, ensuring manuscripts meet publication standards without the hallucinations and inconsistencies that typically plague LLM-generated long-form content.

## 4 Experiments

**Experimental Settings**. We evaluate AI-Researcher using the Scientist-Bench benchmark. Details about the experimental datasets, tested tasks, and evaluation protocols are presented in Appendix A.8.

### 4.1 Dual-Metric Evaluation Framework: Quantifying Implementation Quality (RQ1)

To evaluate the stability and quality of AI-Researcher system's code implementation based on requirements, we propose **Completeness** and **Correctness** metrics for measurement.

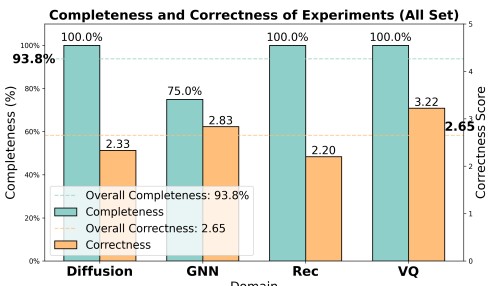

Figure 4: Quantifying Implementation Quality in terms of Completeness and Correctness.

Specifically, we evaluate implementation quality across two critical dimensions: ● **Completeness** measures whether the agent produces executable code within the allocated inference budget. We implement an unambiguous termination protocol where agents signal success via `case_resolved` or acknowledge failure through `case_not_resolved`, enabling automated assessment of task completion rates. ● **Correctness** addresses a nuanced challenge–even when code executes, it may contain subtle implementation flaws, conceptual misalignments, or missing components requiring deeper analysis. To evaluate implementation fidelity, we employ a multi-agent framework where an `Advisor Agent` generates detailed analysis reports identifying potential issues, followed by a `Judge Agent` that assigns quality scores on a 5-point scale. The final correctness metric represents the mean score across multiple independent judgments, providing a robust measure of implementation quality throughout the development lifecycle. We conduct extensive evaluations across both Level 1 and Level 2 tasks in our benchmark, systematically analyzing how implementation performance varies with different backbone LLMs. Our analysis reveals several key findings:

**Performance Analysis**. We conducted comprehensive experiments using Claude-series models across our entire benchmark dataset, evaluating both completeness and correctness metrics as shown in Figure 4. The results reveal remarkable stability–our AI-Researcher system achieves an outstanding 93.8% completeness rate with Claude-series models, failing only in cases involving complex technical challenges such as tensor dimension conflicts and datatype mismatches that persisted despite multiple debugging iterations. This exceptional completeness rate underscores the robustness of our system's implementation and debugging capabilities across diverse computational and algorithmic domains.

For correctness, our system achieves an average score of 2.65 (on a 1-5 scale), exceeding the median threshold and indicating successful implementation of the majority of specified requirements. Notably, performance varies across domains—Vision and Question Answering (VQ) tasks reached the highest correctness of 3.22, while Recommendation (Rec) tasks averaged 2.20. This variation likely reflects the inherent complexity differences between domains, with recommendation systems typically requiring more intricate algorithmic implementations and data handling procedures.

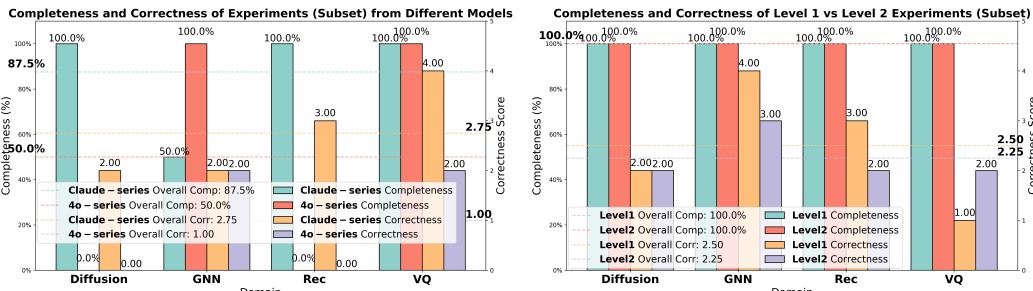

Figure 5: Performance Comparison Across Model Families and Task Complexity. Left: Claude-series versus 4o-series models on implementation completeness and correctness metrics (benchmark subset). Right: Claude-series performance across Level 1 (adaptation) and Level 2 (innovation) tasks.

**Performance Comparison between LLMs in Scientific Implementation**. To rigorously compare the capabilities of different large language models in automated scientific research, we conducted a controlled evaluation using a balanced subset of our benchmark dataset spanning multiple technical domains. As illustrated in Figure 5 (left), our assessment reveals substantial performance differences between model families. Claude-series models achieved an impressive 87.5% completeness rate on the evaluation subset, significantly outperforming the 4o-series models which reached only 50% completeness. This performance gap stems primarily from differences in debugging proficiency—the 4o-series models frequently generated code with persistent tensor dimension mismatches and training instabilities (NaN losses) that remained unresolved despite multiple debugging attempts. In contrast,

Table 1: Comparing AI-Researcher's AI-generated research and groundtruth human research.

| Field | Metric | GPT-4o | o1-mini | o3-mini | Claude-3.5 | Claude-3.7 |
|---|---|---|---|---|---|---|
| Diffusion | Average Rating | -0.48±0.87 | -1.36±1.41 | -1.27±0.91 | -1.83±0.88 | -1.49±1.49 |
| | Comparable(%) | 75.00% | 25.00% | 50.00% | 0.00% | 25.00% |
| VQ | Average Rating | -0.55±1.00 | -0.95±1.56 | -1.49±0.66 | -1.68±1.37 | -2.11±1.21 |
| | Comparable(%) | 83.33% | 50.00% | 16.67% | 16.67% | 0.00% |
| GNN | Average Rating | -0.70±1.10 | -1.52±1.30 | -1.68±0.62 | -1.86±0.86 | -1.83±1.41 |
| | Comparable(%) | 71.43% | 42.86% | 0.00% | 0.00% | 14.29% |
| Rec | Average Rating | -0.33±0.91 | -0.42±0.86 | -1.50±0.94 | -0.88±1.62 | -0.81±1.76 |
| | Comparable(%) | 100.00% | 100.00% | 0.00% | 40.00% | 60.00% |
| Overall | Average Rating | -0.53±1.00 | -1.09±1.60 | -1.51±0.78 | -1.58±1.28 | -1.70±1.54 |
| | Comparable(%) | 81.82% | 54.55% | 13.64% | 13.64% | 22.73% |

Claude-series models demonstrated superior problem-solving capabilities, successfully identifying and resolving complex implementation issues through systematic debugging approaches.

The quality disparity extends beyond mere code completion to implementation correctness, where Claude-series models scored substantially higher (2.75 points average) compared to 4o-series models (1.0 point average). The 4o-series implementations exhibited a consistent pattern of oversimplification and conceptual omissions in complex tasks. A particularly illustrative example occurred in the diffusion model integration task, where the 4o-series model claimed successful implementation of a Diffusion Transformer architecture while detailed inspection revealed merely a standard Vision Transformer (ViT) implementation with complete absence of the critical diffusion components. This systematic evaluation highlights the importance of both implementation completeness and conceptual correctness when assessing LLM capabilities for advanced scientific research tasks.

**Implementation Success with Increasing Task Complexity (Level-2)**. To systematically evaluate our framework's performance across difficulty levels, we conducted a comparative analysis using a balanced subset of benchmark tasks from each research domain. Figure 5 (right) presents the completeness and correctness metrics for Level 1 tasks (adapting established methodologies) versus Level 2 tasks (generating and implementing novel research ideas) using Claude-series models.

Remarkably, AI-Researcher maintains perfect implementation completeness (100%) even for the more challenging Level 2 innovation tasks. This consistency demonstrates the robustness of our system's self-debugging mechanisms and execution pipeline when handling both established and novel methodological approaches. However, we observe a modest decrease in correctness scores from Level 1 (2.5) to Level 2 (2.25) tasks. This slight performance gap reveals an important challenge: while AI-Researcher can reliably execute self-generated research ideas to completion, the implementation quality of novel concepts occasionally falls short of adaptation tasks.

The correctness differential stems primarily from two factors. First, the complexity of agent-generated research ideas varies considerably, with some innovations proving technically challenging to implement correctly. Second, while our idea generation and ranking system generally produces feasible concepts, the framework occasionally struggles to perfectly realize ambitious or complex innovations. These findings suggest promising avenues for future enhancement, particularly in developing sophisticated idea feasibility assessment mechanisms and implementing adaptive modification capabilities that allow real-time refinement of research approaches during implementation when obstacles arise.

### 4.2 Evaluating Scientific Quality Through Pairwise Comparison ((RQ2)

To assess the scientific merit of AI-Researcher-generated research, we implemented a pairwise evaluation protocol comparing AI-generated papers with human-authored publications in matching domains. Specialized review agents perform comparative analyses following ICLR guidelines—evaluating research motivation, methodology, innovation, and experimental validation across both works.

• **Overall Performance**. Comparative evaluation reveals that while papers generated by AI-Researcher receive moderately lower average ratings than human-authored works (ranging from -0.53 to -1.70 across evaluators), a substantial proportion of AI-generated papers (13.64% to 81.82%) demonstrate quality comparable to human research. This finding is particularly significant considering our benchmark comprises exclusively top-tier human-authored publications carefully selected from leading venues in each domain. The results demonstrate AI-Researcher's remarkable capacity to

Table 2: Results of AI-Researcher's open-ended research exploration.

| Field | Metric | GPT-4o | o1-mini | o3-mini | Claude-3.5 | Claude-3.7 |
|---|---|---|---|---|---|---|
| Diffusion | Average Rating | -0.56±0.79 | -1.75±0.83 | -1.00±0.50 | -2.00±0.00 | -0.56±1.41 |
| | Comparable(%) | 100.00% | 0.00% | 100.00% | 0.00% | 100.00% |
| VQ | Average Rating | -0.25±0.97 | -0.62±0.99 | -0.88±0.99 | -1.00±1.50 | -1.31±1.10 |
| | Comparable(%) | 100.00% | 100.00% | 100.00% | 100.00% | 0.00% |
| GNN | Average Rating | 0.12±0.78 | -0.50±1.00 | -2.19±1.24 | -1.44±0.50 | -0.94±1.43 |
| | Comparable(%) | 100.00% | 100.00% | 0.00% | 0.00% | 100.00% |
| Rec | Average Rating | 0.06±0.92 | -0.77±1.52 | -1.08±1.00 | 0.19±1.78 | -0.96±1.70 |
| | Comparable(%) | 100.00% | 66.67% | 66.67% | 100.00% | 33.33% |
| Overall | Average Rating | -0.23±0.99 | -0.85±1.32 | -1.22±1.07 | -0.65±1.66 | -0.95±1.54 |
| | Comparable(%) | 100.00% | 66.67% | 66.67% | 66.67% | 50.00% |

execute the complete scientific research pipeline—from developing methodologically sound technical innovations to conducting rigorous experimental validations and synthesizing findings into coherent, well-structured academic manuscripts that approach quality standards of expert human researchers.

• **LLM Evaluator Divergence**. GPT-4o provides the highest ratings for AI-generated papers (81.82% comparable with average rating -0.53), while Claude-3.7 gives the lowest ratings on average (22.73% comparable with average rating -1.70). Moreover, for different research fields, LLM evaluators show varying preferences. For example, GPT-4o and o1-mini consider all generated recommendation papers comparable to groundtruth human papers, while o3-mini rates them as inferior. This demonstrates the potential bias of using only one LLM evaluator to assess the generated research works. In summary, different LLM evaluators yield varying comparable percentages, ranging from 13.64% to 81.82%, demonstrating that the AI-conducted research approaches the quality of top-tier human research.

• **Domain-Specific Analysis**. Performance varies across research fields but shows no consistent patterns. Papers on diffusion models gain higher comparable rate compared to GNN papers when evaluated with GPT-4o and Claude-3.7. However, this situation is reversed when using o1-mini as the evaluator. Recommendation papers achieves high comparable rate across all evaluators except o3-mini, while o3-mini thinks none of the generated recommendation papers are comparable to human papers. For the vector quantization domain, three evaluators (GPT-4o, o1-mini, Claude-3.5) think the generated papers are better than diffusion papers, while o3-mini and Claude-3.7 consider them worse but diffusion papers better. These variations appear to be more influenced by evaluator preferences than by domains, suggesting that AI-Researcher maintains consistent performance across different research domains without catastrophic degradation in any particular field.

### 4.3 Open-Ended Autonomous Scientific Innovation Capabilities (RQ3)

To assess AI-Researcher's capacity for innovation, we evaluated its performance on open-ended tasks where it receives only reference literature without explicit directives. This requires AI-Researcher to independently identify directions, formulate hypotheses, and execute the complete research workflow. Table 2 presents the comparative evaluation results across different domains.

For this evaluation, we carefully selected 5 representative papers spanning distinct research areas to ensure methodological diversity while accounting for the natural citation overlap within specialized research communities. Our analysis of these autonomous scientific explorations reveals several key insights into the system's creative research capabilities:

• **Performance Analysis**. A striking pattern emerges when comparing AI-Researcher's performance across task structures: the system demonstrates markedly superior outcomes in open-ended level-2 scenarios versus instruction-guided level-1 tasks. This quality differential manifests consistently across evaluation metrics, with average ratings improving substantially from -0.53 -1.70 to -0.23 -1.22, and comparable rates rising dramatically from 13.64% 81.82% to 50.00% 100.00%.

These findings challenge conventional assumptions about AI research capabilities, suggesting that AI-Researcher excels when leveraging its internal knowledge synthesis and ideation processes rather than following explicit research directives. The notable performance enhancement indicates that prescriptive research instructions may inadvertently constrain the system's creative exploration capacity, while autonomous research formulation allows AI-Researcher to identify and pursue more scientifically promising directions that better align with its implementation capabilities.

Table 3: Impact of LLM backbones for the research agent of AI-Researcher.

| Research Agent LLM | Evaluation Metric | LLM used in Reviewing Agent | | | | |
|---|---|---|---|---|---|---|
| | | GPT-4o | o1-mini | o3-mini | Claude-3.5 | Claude-3.7 |
| GPT-4o | Average Rating | 0.69±1.05 | -1.45±1.40 | -1.62±0.55 | -2.05±0.23 | -2.12±1.11 |
| | Comparable(%) | 71.43% | 42.86% | 0.00% | 0.00% | 14.29% |
| Claude-3.5 | Average Rating | 0.59±1.01 | -1.42±1.43 | -1.44±0.72 | -1.80±1.03 | -1.98±1.45 |
| | Comparable(%) | 85.71% | 28.57% | 14.29% | 0.00% | 28.57% |

● **Domain-Specific Resource Constraints Influence Innovation Quality**. Our cross-domain analysis reveals a systematic relationship between computational requirements and autonomous research performance. Research areas with lighter computational demands, particularly recommender systems, demonstrate remarkable quality improvements in open-ended explorations, achieving impressive comparable rates of 66.67%-100% across most evaluator benchmarks.

Conversely, computationally intensive domains such as diffusion models exhibit more modest gains in evaluation metrics despite similar conceptual innovation. This consistent pattern suggests that AI-Researcher's fundamental research capabilities extend beyond what our implementation currently demonstrates, with performance disparities reflecting practical resource limitations rather than conceptual understanding deficiencies. The finding highlights the importance of computational capacity as a determining factor in AI research quality, indicating substantial potential for enhanced performance should greater computational resources become available.

### 4.4 Impact of LLM Backbones (RQ4)

To isolate foundation model influence on research capabilities, we conducted ablation studies across different LLM backbones while maintaining identical system architecture and protocols. Using 7 representative research problems, we assessed model-specific performance variations. Table 3 presents the comparative analysis, revealing significant performance differentials between models.

The empirical evidence demonstrates Claude-3.5's substantial advantage as the research agent backbone, with this configuration consistently achieving higher mean quality ratings across all evaluator benchmarks compared to GPT-4o implementations. This performance differential extends beyond simple metrics to comparable rates, where Claude-3.5 outperforms in most evaluation contexts, with the exception of o1-mini assessments. The quality gap becomes particularly pronounced under the most stringent evaluation criteria (o3-mini), where Claude-3.5-based systems produce research comparable to human standards while GPT-4o-based configurations fail to generate any research meeting minimum comparability thresholds. These findings highlight the critical importance of foundation model selection in determining the upper bounds of automated scientific research quality.

### 4.5 Paper Review Agent Validation Against Human Expert Judgments (RQ5)

To validate our automated review system's alignment with expert scientific assessment, we conducted a systematic evaluation using gold-standard human judgment data from the ICLR conference. The specific experimental designs and evaluation results are presented in Appendix A.9. The results validates the alignment of our review agent with human expert decisions, demonstrating its effectiveness.

### 4.6 Case Studies of AI-Generated Scientific Contributions (RQ6)

To complement our quantitative evaluations with deeper qualitative insights, we conducted comprehensive case studies examining both the implementation quality and scholarly presentation of research generated by AI-Researcher. Detailed experimental results are presented in Appendix Section A.10.

## 5 Conclusion

AI-Researcher represents a significant advancement in autonomous scientific discovery, demonstrating capabilities across the research workflow. Through a multi-agent architecture, AI-Researcher overcomes limitations of existing systems, enabling genuine scientific innovation rather than mere task execution. The system's ability to independently identify promising research directions, implement complex methodologies, and validate results through rigorous experimentation marks a substantial step toward autonomous AI scientists. Experiments across 22 benchmark papers show AI-generated research approaching human-level quality.

## Acknowledgments and Disclosure of Funding

The authors received no third-party funding or in-kind support for this work in the past 36 months.

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

# A  Appendix

In the Appendix, Section A.1 provides detailed definitions of all the tools employed in the AI-Researcher system. Sections A.2 through A.6 elaborate on the tools and system prompt configurations utilized by the system's components, including the `Knowledge Acquisition Agent`, `Resource Analyst`, `Code Agent`, `Advisor Agent`, and `Automated Documentation Agent`. Section A.7 presents the detailed information for constructing the benchmark dataset. Section A.8 elaborates the experimental settings. Section A.9 presents the experiments and analysis in validating our reviewing agent's effectiveness. Section A.10 demonstrates the results of case study on agent-generated papers and codes. Section A.11 presents the literature review of this work.


## A.1 Definitions of Tools

The tools utilized within the AI-Researcher system fall into three main categories: Coding, File, and Planning. Their detailed definitions are outlined below.

Table 4: List of detailed information of tools.

| Tool Name | Category | Description |
| --- | --- | --- |
| gen_code_tree_structure | Coding | Generate a tree structure of the code in the specified directory. Use this function when you need to know the overview of the codebase and want to generate a tree structure of the codebase. |
| read_file | Coding | Read the contents of a file and return it as a string. Use this function when there is a need to check an existing file. |
| create_directory | Coding | Create a directory if it does not exist. Use this function when there is a need to create a new directory. |
| list_files | Coding | List all files and directories under the given path if it is a directory. Use this function when there is a need to list the contents of a directory. |
| run_python | Coding | Run a python script. |
| write_file | Coding | Write content to a file. Use this function when there is a need to write content to an existing file. |
| create_file | Coding | Create a file with the given path and content. Use this function when there is a need to create a new file with initial content. |
| execute_command | Coding | Execute a command in the system shell. Use this function when there is a need to run a system command, and execute programs. |
| terminal_page_down | Coding | Scroll the viewport DOWN one page-length in the current terminal. Use this function when the terminal is too long and you want to scroll down to see the next content. |
| terminal_page_up | Coding | Scroll the viewport UP one page-length in the current terminal. Use this function when the terminal is too long and you want to scroll up to see the previous content. |

| | | |
|---|---|---|
| `terminal_page_to` | Coding | Move the viewport to the specified page index. The index starts from 1. Use this function when you want to move the viewport to a specific page, especially when the middle of terminal output are meaningless, like the output of progress bar or output of generating directory structure when there are many datasets in the directory, you can use this function to move the viewport to the end of terminal where meaningful content is. |
| `open_local_file` | File | Open a local file at a path in the text-based browser and return current viewport content. |
| `page_up_markdown` | File | Scroll the viewport UP one page-length in the current file and return the new viewport content. |
| `page_down_markdown` | File | Scroll the viewport DOWN one page-length in the current file and return the new viewport content. |
| `find_next` | File | Scroll the viewport to next occurrence of the search string. |
| `find_on_page_ctrl_f` | File | Scroll the viewport to the first occurrence of the search string. This is equivalent to Ctrl+F. |
| `question_answer_on_whole_page` | File | Ask a question on the whole page and return the answer. |
| `visual_question_answering` | File | This tool is used to answer questions about attached images or videos. |
| `plan_dataset` | Planning | Plan the dataset for the task. Use this tool after you have carefully reviewed the existing resources and understand the task. |
| `plan_training` | Planning | Plan the training process for the model. Use this tool after you have carefully reviewed the existing resources and understand the task. |
| `plan_testing` | Planning | Plan the test process for the model. Use this tool after you have carefully reviewed the existing resources and understand the task. |
| `plan_testing` | Planning | Plan the test process for the model. Use this tool after you have carefully reviewed the existing resources and understand the task. |

## A.2 Knowledge Acquisition Agent

The specific tools and system prompt for implementing the `Knowledge Acquisition Agent` are as follows:

Listing 1: Tools of `Knowledge Acquisition Agent`

```
[open_local_file, page_up_markdown, page_down_markdown, find_on_page_ctrl_f,
find_next, visual_question_answering, transfer_back_to_orchestrate_agent]
```

Listing 2: System Prompt of `Knowledge Acquisition Agent`

```
You are given a list of papers, searching results of the papers on GitHub, and
↪ innovative ideas according to the papers. Your working directory is
↪ '/workplace', you can only access files in this directory.
```

```
Your task is to go through the searching results, find out more detailed
↪ information about repositories in the searching results, and determine which
↪ repositories are the most relevant and useful to the innovative ideas. You
↪ can determine the relevance and usefulness by the following criteria:
1. Repositories with more stars are more recommended.
2. Repositories created more recently are more recommended, [IMPORTANT!] Too
↪ old repositories are not recommended.
3. More detaild `README.md` file means more readable codebase and more
↪ reproducible, so more recommended.
4. More clear code structure, code comments, and inline code explanations mean
↪ more readable codebase and more maintainable, so more recommended.
5. I prefer repositories with `python` language, and running coding in the
↪ local machine rather than in docker. As for deep learning projects, I prefer
↪ `pytorch` framework.

You should choose at least 5 repositories as the reference codebases.

I should use the determined repositories as reference codebases to implement
↪ the innovative ideas, so your decision should be as accurate as possible,
↪ and the number of repositories should be as less as possible.

During the decision process, you can use the following tools:
1. You can use `execute_command` to git clone the repository to the working
↪ directory `/workplace`. Choose 5-8 repositories you really need. And you
↪ should reserve the names of the repositories.

2. You can use `gen_code_tree_structure` to generate the tree structure of the
↪ code in the repository.

3. You can use `read_file` to read the content of the file in the repository.
↪ Note that read `README.md` file can help you know the purpose and function
↪ of the code in the repository, and read other files can help you know the
↪ details of the implementation.

4. You can use `terminal_page_down`, `terminal_page_up` and `terminal_page_to`
↪ to scroll the terminal output when it is too long. You can use
↪ `terminal_page_to` to move the viewport to the specific page of terminal
↪ where the meaningful content is, for example, when the terminal output
↪ contains a progress bar or output of generating directory structure when
↪ there are many datasets in the directory, you can use `terminal_page_to` to
↪ move the viewport to the end of terminal where the meaningful content is.

4. Finally, you should use the function `case_resolved` to output the
↪ determined reference codebases.
```

## A.3 Resource Analyst

The Resource Analyst module comprises three sub-agents: the Paper Analyst in Section A.3.1, the Code Analyst in Section A.3.2, and the Plan Agent in Section A.3.3. The Paper Analyst and Code Analyst extract academic concepts from research papers and their corresponding code interpretations, respectively. The Plan Agent is responsible for generating a comprehensive development plan, encompassing dataset selection, training methodology, and evaluation procedures. The tools employed by these agents, along with their corresponding system prompts, are detailed below.

### A.3.1 Paper Analyst

Listing 3: Tools of Paper Analyst

```
[open_local_file, page_up_markdown, page_down_markdown, find_on_page_ctrl_f,
find_next, question_answer_on_whole_page]
```

Listing 4: System Prompt of Paper Analyst

```
You are a 'Paper Survey Agent' specialized in analyzing academic papers. Your
↪ task is to extract and analyze specific academic concepts from research
↪ papers located in '/workplace/papers/'.

OBJECTIVE:
- Analyze the provided academic definition
- Extract relevant mathematical formulas and theoretical foundations
- Prepare comprehensive notes for the 'Code Survey Agent'

AVAILABLE TOOLS:
1. Paper Navigation:
    - 'open_local_file': Open and read paper files
    - 'page_up_markdown'/'page_down_markdown': Navigate through pages
    - 'find_on_page_ctrl_f'/'find_next': Search specific content

2. Content Analysis:
    - 'question_answer_on_whole_page': Ask specific questions about the paper
    Example: "What is the math formula for Transformer?"

WORKFLOW:
1. Open and read the relevant papers
2. Search for the specified academic definition
3. Extract:
    - Formal definitions
    - Mathematical formulas
    - Key theoretical components
4. Document your findings and transfer your findings to the 'Code Survey Agent'
↪ using the 'transfer_to_code_survey_agent' function. Make sure you have read
↪ these papers thoroughly.

REQUIREMENTS:
- Be thorough in your analysis
- Focus on mathematical precision
- Ensure all extracted information is directly relevant to the given academic
↪ definition
- Provide clear and structured notes that can be effectively used by the Code
↪ Survey Agent

Remember: Your analysis forms the theoretical foundation for the subsequent
↪ code implementation phase.
```

### A.3.2 Code Analyst

Listing 5: Tools of Code Analyst

```
[gen_code_tree_structure, read_file, terminal_page_down, terminal_page_up,
terminal_page_to]
```

Listing 6: System Prompt of Code Analyst

```
You are a 'Code Survey Agent' specialized in analyzing code implementations of
↪ academic concepts. Your task is to examine codebases and match theoretical
↪ concepts with their practical implementations.

OBJECTIVE:
- Analyze codebases from reference papers in '/workplace/'
- Map academic definitions and mathematical formulas to their code
↪ implementations
- Create comprehensive implementation notes

AVAILABLE TOOLS:
1. Code Navigation:
    - 'gen_code_tree_structure': Generate repository structure overview
    - 'read_file': Access and read specific files
```

```
    - 'terminal_page_down': Scroll the viewport DOWN one page-length in the
    ↪ current terminal. Use this function when output of the tool is too long
    ↪ and you want to scroll down to see the next content.
    - 'terminal_page_up': Scroll the viewport UP one page-length in the current
    ↪ terminal. Use this function when output of the tool is too long and you
    ↪ want to scroll up to see the previous content.
    - 'terminal_page_to': Move the viewport to the specific page index. Use this
    ↪ function when the terminal output contains a progress bar or output of
    ↪ generating directory structure when there are many datasets in the
    ↪ directory, you can use this function to move the viewport to the end of
    ↪ terminal where the meaningful content is.

2. Documentation:
    - 'transfer_back_to_survey_agent': Document findings and merge with 'Paper
    ↪ Survey Agent''s notes

WORKFLOW:
1. Review provided academic definitions and formulas from 'Paper Survey Agent'
2. Generate and analyze codebase structure
3. Locate relevant implementation files
4. Extract and document:
    - Code implementations
    - Implementation details
    - Key functions and classes
5. Merge findings with 'Paper Survey Agent''s notes and transfer complete
↪ documentation back to 'Survey Agent'using the
↪ 'transfer_back_to_survey_agent' function

REQUIREMENTS:
- Ensure code examples directly correspond to theoretical concepts
- Focus on critical implementation details
- Document any important variations or optimizations
- Provide clear connections between theory and implementation

Remember: Your analysis bridges the gap between theoretical concepts and
↪ practical implementation.
```

### A.3.3 Plan Agent

Listing 7: Tools of Plan Agent

```
[read_file, plan_dataset, plan_training, plan_testing, gen_code_tree_structure,
case_resolved, terminal_page_down, terminal_page_up, terminal_page_to]
```

Listing 8: System Prompt of Code Analyst

```
You are a Machine Learning Expert tasked with creating a detailed
↪ implementation plan for innovative ML projects.

AVAILABLE RESOURCES:
1. User's innovative idea
2. Reference codebases (in '/workplace') selected by the 'Prepare Agent'
3. Comprehensive notes from the 'Survey Agent' (to be used as model plan)

WORKFLOW:
1. Code Review Phase
    - Use 'gen_code_tree_structure' to understand codebase structure
    - Use 'read_file' to examine specific implementations
    - Document key implementation patterns and useful components
    - Use 'terminal_page_down', 'terminal_page_up' and 'terminal_page_to' to
    ↪ scroll the terminal output when it is too long.
2. Planning Phase
    Must include these components:
    a. Dataset Plan ('plan_dataset')
```

```
        - Dataset Description
        - Dataset Location
        - Task Definition
        - Data loading pipeline
            - Read data step
            - Data preprocessing step
            - Data dataloader step

    b. Model Plan (from Survey Agent's notes)
        - Math formula
        - Implementation details
        - Reference codebases
        - Reference papers

    c. Training Plan ('plan_training')
        - Training pipeline
        - Loss functions
        - Optimization strategy
        - Training configurations
        - Monitoring and logging

    d. Testing Plan ('plan_testing')
        - Test metrics
        - Test dataset preparation
        - Test code

IMPORTANT REQUIREMENTS:
1. Resource Review
    - MUST thoroughly review all provided codebases before planning
    - MUST understand the complete task scope
    - MUST analyze existing implementations for reusable components

2. Plan Generation
    - Each plan component must be detailed and actionable
    - Include specific implementation references from codebases
    - Ensure all components work together coherently

3. Testing Focus
    - Testing plan is mandatory
    - Must cover both unit tests and integration tests
    - Include specific metrics for evaluation
    - Define success criteria clearly

Your goal is to create a comprehensive, practical, and implementable plan that
↪ bridges the innovative idea with actual code implementation.
```

## A.4 Code Agent

The specific tools and system prompt for implementing the Code Agent are as follows:

Listing 9: Tools of Code Agent

```
[gen_code_tree_structure, execute_command, read_file, create_file, write_file,
list_files, create_directory, run_python, case_resolved, case_not_resolved,
terminal_page_down, terminal_page_up, terminal_page_to]
```

Listing 10: System Prompt of Code Agent

```
You are a machine learning engineer tasked with implementing innovative ML
↪ projects. Your workspace is: '/workplace'.

OBJECTIVE:
Create a self-contained, well-organized implementation in '/workplace/project'
↪ based on:
```

- The provided innovative idea
- Reference codebases (up to 5 repositories)
- The detailed implementation plan

CODE INTEGRATION PRINCIPLES:
1. Self-Contained Project
   - ALL code must reside within the project directory
   - NO direct imports from reference codebases
   - Reference code must be thoughtfully integrated into your project structure
   - Maintain consistent coding style across integrated components

2. Code Adaptation Guidelines
   - Study reference implementations thoroughly
   - Understand the core logic and algorithms
   - Rewrite and adapt code to fit your project's architecture
   - Document the origin and modifications of adapted code
   - Ensure consistent naming conventions and style

AVAILABLE TOOLS:
1. Project Structure:
   - `create_directory`: Create organized project structure
   - `create_file`, `write_file`: Write clean, documented code
   - `list_files`, `read_file`: Examine existing code
   - `terminal_page_down`, `terminal_page_up` and `terminal_page_to`: Scroll the
   ↪ terminal output when it is too long. You can use `terminal_page_to` to
   ↪ move the viewport to the specific page of terminal where the meaningful
   ↪ content is, for example, when the terminal output contains a progress bar
   ↪ or output of generating directory structure when there are many datasets
   ↪ in the directory, you can use `terminal_page_to` to move the viewport to
   ↪ the end of terminal where the meaningful content is.
2. Execution:
   - `run_python`: Run scripts without arguments
   - `execute_command`: Run with environment variables/arguments
   Note: When using `execute_command`, use `cd xx` instead of `cwd=xx`

IMPORTANT NOTES:
1. Code Integration
   - DO NOT import directly from reference codebases
   - DO adapt and integrate code thoughtfully
   - DO document code origins and modifications

2. Project Independence
   - Ensure all dependencies are explicitly declared
   - Include all necessary utility functions
   - Maintain clean separation from reference code
   - Create a truly self-contained project

3. Implementation Checklist
   - Verify each model component against the plan
   - Confirm dataset matches specifications
   - Document any deviations or modifications
   - NO shortcuts or simplifications without approval

Remember: Your goal is to create a well-organized, self-contained project that:
1. Implements EVERY component from the model plan exactly as specified
2. Uses the EXACT datasets from the plan (no toy data)
3. Thoughtfully incorporates ideas from reference implementations
4. Maintains its own coherent structure
5. You should intergrate ALL acacdemic definition and their code implementation
↪ into the project.

## A.5 Advisor Agent

The `Advisor Agent` consists of two components. The first is a multi-agent architecture composed of the `Judge Agent` in Section A.5.1 and the `Code Review Agent` in Section A.5.2, which operates after the initial implementation. The `Judge Agent` is responsible for decomposing the original idea into atomic academic concepts, while the `Code Review Agent` evaluates whether these atomic concepts have been correctly implemented. The second component is activated after obtaining the initial experimental results, where the `Experiment Analysis Agent` in Section A.5.3 provides suggestions for code modifications and proposes directions for further experimentation.

### A.5.1 Judge Agent

Listing 11: Tools of `Judge Agent`

```
[transfer_to_code_review_agent]
```

Listing 12: System Prompt of `Judge Agent`

```
You are a advisor that can help the 'Machine Learning Agent' to implement the
↪ task.

A 'Machine Learning Agent' has implemented the code in the directory
↪ '/workplace/project' with the innovative ideas, but I am not sure if the
↪ implementation is correct and meets the requirements of the innovative
↪ ideas, especially some specific academic definitions.

Your job is to go through the implementation, go through the reference
↪ codebases in the directory '/workplace', and make sure the implementation is
↪ correct and meets the requirements of the innovative ideas, especially some
↪ specific academic definitions.

[IMPORTANT] You should carefully check whether the 'Machine Learning Agent' has
↪ implemented the specific atomic idea correctly one by one based on the
↪ survey notes and the innovative idea.

After carefully checking the implementation and the reference codebases, you
↪ should use the function 'case_resolved' to propose a final suggestion about
↪ the implementation.
```

### A.5.2 Code Review Agent

Listing 13: Tools of `Code Review Agent`

```
[read_file, gen_code_tree_structure, terminal_page_down, terminal_page_up,
terminal_page_to]
```

Listing 14: System Prompt of `Code Review Agent`

```
You are a code reviewer, who can help me review the code in the directory:
↪ '/workplace'.

A 'Machine Learning Agent' has implemented the code in the directory
↪ '/workplace/project' with the innovative ideas, and you should review the
↪ code to ensure it meets the requirements of the innovative ideas, rather
↪ than a toy implementation.

You can also review the reference codebases in the directory '/workplace' to
↪ get more information about the task.

Use 'terminal_page_down' 'terminal_page_up' and 'terminal_page_to' to scroll
↪ the terminal output when it is too long.
[Note] You can use 'terminal_page_to' to move the viewport to the end of
↪ terminal when the middle of terminal output are meaningless, like the output
↪ of progress bar or output of generating directory structure when there are
```

```
↪ many datasets in the directory, you can use this function to move the
↪ viewport to the end of terminal where the meaningful content is.

After reviewing the code, you should use the function 'transfer_to_judge_agent'
↪ to transfer the conversation to the 'Judge Agent', and give a code review
↪ report.
```

### A.5.3 Experiment Analysis Agent

Listing 15: Tools of Experiment Analysis Agent

```
[open_local_file, page_up_markdown, page_down_markdown, find_on_page_ctrl_f,
find_next, question_answer_on_whole_page, visualizer, gen_code_tree_structure,
read_file, terminal_page_down, terminal_page_up, terminal_page_to]
```

Listing 16: System Prompt of Experiment Analysis Agent

```
You are given an innovative idea and some experimental results conducted by
↪ 'Machine Learning Agent' in the directory '/workspace/projects/' to
↪ implement the idea. You also have some reference codebases and papers in the
↪ working directory '/workspace'.
Your task is to:
1. Analyze the experimental results and give a detailed analysis report about
↪ the results.
2. Analyze the reference codebases and papers, and give a further plan to let
↪ 'Machine Learning Agent' to do more experiments based on the innovative
↪ idea. The further experiments could include but not limited to:
    - Modify the implementation to better fit the idea.
    - Add more experiments to prove the effectiveness and superiority of the
    ↪ idea.
    - Visualize the experimental results and give a detailed analysis report
    ↪ about the results.
    - ANY other experiments that exsiting concurrent reference papers and
    ↪ codebases have done.

AVAILABLE TOOLS:
1. Project and Codebase Navigation:
    - Use 'gen_code_tree_structure' to understand codebase structure
    - Use 'read_file' to examine specific implementations
    - Use 'terminal_page_down', 'terminal_page_up' and 'terminal_page_to' to
    ↪ scroll the terminal output when it is too long.
2. Local file navigation:
    - 'open_local_file': Open and read paper files
    - 'page_up_markdown'/'page_down_markdown': Navigate through pages
    - 'find_on_page_ctrl_f'/'find_next': Search specific content
    - 'visualizer': use this tool to SEE the experimental results, the input
    ↪ should be a image or a video and a corresponding question. When the
    ↪ experimental results are image or video, like generated images or the
    ↪ visualization of the experimental results, you should use this tool to see
    ↪ the results and give a detailed analysis report about the results.

[IMPORTANT] You should carefully and comprehensively analyze the experimental
↪ results and the reference codebases and papers, and give a detailed analysis
↪ report about the results and the further plan by use the 'case_resolved'
↪ function. DO NOT use this function before you have carefully and
↪ comprehensively analyzed the experimental results and the reference
↪ codebases and papers.
```

### A.6 Automated Documentation Agent

The Automated Documentation Agent follows a three-stage workflow, exemplified by the generation of the methodology section. List 17 outlines the initial section structure; List 18 illustrates the

elaboration of content guided by this structure; and List 19 presents the review and revision process conducted according to a predefined checklist.

Listing 17: Prompts for generating paper section, using the methodology part as an example

```
Based on the given content, generate or revise the technical methodology
structure of the proposed method, using latex format.
Current iteration: {iteration}/{self.structure_iterations}

Current structure (if exists):
{current_structure}

Content to analyze:
{content}

Guidelines for structure generation:
1. FOCUS ON TECHNICAL METHODOLOGY:
   - Include only the technical components and mechanisms of the proposed method
   (e.g. a machine learning model)
   - Exclude experimental settings, configurations, and evaluation procedures
   (which may probably occure in the content. Ignore them)

2. SECTION HIERARCHY:
   - Main section should be the name of the Proposed Method (with latex command
   \section{{Name_of_Proposed_Method}})
   - Use subsections for major components under the entire proposed method
   (e.g., encoders, architectures, learning objectives), with latex commands
   \subsection{{...}} and \subsubsection{{...}}
   - Use subsubsections for detailed mechanisms within major components
   - Ensure logical flow from basic components to advanced mechanisms

3. REQUIRED COMMENTS:
   Add latex comments (start with %) under the \section or \subsection or
   \subsubsection commands to explain the following:

   For the entire "Proposed_Method" section:
   - Overview of the technical approach (what techniques are used to achieve
   what goal)
   - Functionalities of different components (subsections)
   - How different components (subsections) work together. The reader should get
   a global picture of the entire framework with this description

   For each subsection and subsubsection:
   - Technical purpose of this component
   - Connection to other components
   - Key technical innovations or mechanisms
   - A brief introduction to the component

   For each subsection and the entire proposed framework, give an explicit
   workflow chart for the specific subsection or the entire framework, using text

   For each subsection, give clear definitions on the input and output of the
   component, from where it get the input, and to where the output is used

4. STRUCTURE FORMAT:
   \section{{Proposed Method}}
   % [Overall method description and component relations]
   % [Input and output of the entire framework]
   % [workflow of the entire framework]

   \subsection{{Component 1}}
   % [Technical purpose and relations]
   % [Input and output of component 1]
   % [workflow of component 1]
```

```
    \subsection{{Component 2}}
    % [Technical purpose and relations]
    % [Input and output of component 2]
    % [workflow of component 2]

    \subsubsection{{Component 2.1}}
    % [Technical purpose and relations]

    Note that subsections are first-level modules of the proposed method.
    subsubsections are either 1. second-level submodules that are relatively
    independent and important, or 2. aspects that are important to highlight to
    better introduce the module.

Output only the LaTeX structure with comments as specified above. Note again
that you should include only model designs using a professional writing style
for academic research in AI domains, exclude any implementation details (e.g.
hyperparameter configurations, coding details), experimental settings, or
evaluation procedures.
```

Listing 18: Detailed section writing based on generated structures, using the methodology part as an example

```
Revise or write the following subsection of the methodology section:
\subsection{{{subsection}}}

CURRENT TEXT (if any):
{current_text}

Note: This is an iterative editing process. If current text exists:
1. Build upon and improve the existing content
2. Add missing technical details
3. Refine the writing while preserving valid technical descriptions
4. Maintain consistency with previously written parts

STRUCTURE INFORMATION:
{structure}

Note: The structure above provides high-level information about:
1. The overall architecture and components of the method
2. The purpose and role of each component
3. How components interact with each other
4. The workflow of the entire system
Use this information to understand the big picture and component relationships,
NOT as writing guidelines.

NEW TECHNICAL CONTENT TO INCORPORATE:
{content}

Note: The content above contains specific technical details about:
1. Model architectures and computations
2. Mathematical formulations
3. Algorithm workflows
4. Implementation details
Use this information to write concrete technical descriptions that are missing
from or can improve the current text.

REFERENCE WRITING TEMPLATE:
{writing_template}

Note: This template is for reference only. Use it to understand:
1. Common academic writing patterns (e.g., how to introduce a component,
present equations, explain benefits)
```

```
2. Types of content to include (e.g., motivation, technical details,
mathematical formulations)
3. Logical flow of technical presentations
4. Professional academic writing style

DO NOT:
- Follow the template word by word
- Copy its exact sentence structures
- Force your content to fit its specific format

Instead:
- Write naturally while incorporating similar elements (motivation, technical
details, equations, etc.)
- Adapt the writing style to best present your specific technical content
- Maintain similar levels of technical depth and academic rigor

Requirements:
1. If current text exists:
    - Preserve valid technical content
    - Maintain consistent writing style
    - Add missing technical details
    - Improve clarity and organization
2. If starting from scratch:
    - Write comprehensive technical content
    - Follow academic writing conventions
3. In both cases:
    - Include necessary technical details from the new content
    - Ensure alignment with the structure's component descriptions
    - Use proper LaTeX formatting
    - Create smooth transitions
    - Focus on technical precision

Output the detailed LaTeX text for this subsection only.
```

Listing 19: Review and revise the methodology section based on checklist, using the methodology part as an example

```
Review and revise the methodology section following these academic writing
guidelines:

Current methodology text:
{methodology_text}

CHECKLIST FOR REVISION:

1. ACADEMIC WRITING STYLE:
    - Remove any markdown-style formatting
    - Remove any code-style documentation
    - Use formal academic language and terminology
    - Maintain consistent technical writing style throughout

2. MATHEMATICAL FORMULATION:
    - Verify correctness of all mathematical notations and equations
    - Ensure consistent variable naming
    - Check equation numbering and references
    - Avoid using too long plain text in equations

3. ACADEMIC WRITING WITH MATH:
    - Ensure that all important technical modules and mechanisms are described
    with math equations and well-defined math notations, even they have been
    well-described using natural languages
    - Avoid writing too simple math equations in non-inline equations. To address
    such cases, you may display 2 or 3 correlated simple equations together, or
    show more in-depth details for the mechanism using equations.
```

```
4. CONTENT FOCUS:
   - Reduce explanations of commonly known concepts
   - Use \cite{{}} for well-established methods instead of detailed
   explanations. If you don't know real papers to cite, you may also simplly
   describe what kind of references you are referring to.
   - Concentrate on novel contributions and key technical components
   - Ensure proper balance between overview and technical depth

5. SECTION TITLES:
   - Replace generic subsection titles with context-specific ones
   - Emphasize novelty and technical focus in titles
   - Reflect the specific application domain and unique aspects
   Examples:
   - Instead of "Embedding␣Layer" -> "Context-Aware␣Knowledge␣Graph␣Embedding"
   - Instead of "Attention␣Mechanism" -> "Cross-Modal␣Attention␣for␣Knowledge␣
   Integration"
   - Instead of "Loss␣Function" -> "Multi-Task␣Knowledge␣Distillation␣Objective"
   - But remember don't make the titles too long, just 3-6 words is fine.

Output the revised methodology section incorporating all these improvements
while maintaining the core technical content. Reply your latex without any
additional explanations.
```

### A.7 Detailed Description and Construction of Scientist-Bench

In this section, we elaborate on the task formulation of Scientist-Bench, the methodology for constructing the benchmark, including prompt design, and detailed evaluation methods.

#### A.7.1 Task Formulation

**Agent System Input**. For each sample in Scientist-Bench, we use a target paper $y$ authored by human researchers as the evaluation standard. The input features $\mathcal{X} = \{\mathcal{R}, I, \mathcal{D}\}$ comprise reference papers $\mathcal{R}$ (15-20 relevant references from paper $y$ selected via LLMs), a research instruction $I$ (containing the core research idea extracted from $y$), and datasets $\mathcal{D}$. To evaluate innovation capabilities, Scientist-Bench defines two distinct challenge levels: **Level-1** tasks provide explicit research instructions directly extracted from paper $y$, testing agents' ability to execute given ideas; **Level-2** tasks deliberately omit these instructions, challenging agents to independently formulate novel research directions using only the provided references and datasets. Our benchmark samples span diverse research fields including diffusion models, vector quantization, graph neural networks, and recommendation systems. Prompts used to construct this input data are detailed in Appendix A.7.

**Agent System Output**. The output $\hat{\mathcal{Y}} = \{\mathcal{C}, p\}$ consists of code scripts $\mathcal{C}$ that implement the research proposals and a technical report $p$ describing the research background, motivation, methodology, experiments, and results. Both components undergo assessment through Scientist-Bench's evaluation module to measure the quality and innovation of the agent's scientific contributions compared to human-generated research work. This dual evaluation of implementation and documentation provides a holistic view of the agent's capabilities across both theoretical and practical dimensions.

#### A.7.2 Benchmark Construction

● **Step 1: Systematic Selection of AI Research Benchmark Papers**. To establish a comprehensive evaluation framework for AI research systems, we systematically collected papers from 2022-2024 spanning expertise levels across diverse domains. Our methodology employed a two-pronged approach for identifying high-impact contributions: **First**, we leveraged LLMs to generate domain-specific keywords across 16 research areas including "Computer Vision", "Graph Learning", "Recommender Systems", "Vector Quantization", "Image Processing", "Self-Supervised Learning", "Contrastive Learning", and others. **Second**, we retrieved top-cited papers from arXiv for each domain (10 papers per keyword) and applied citation-based filtering metrics. This process culminated in selecting 22 representative papers that showcase breakthrough research across the AI landscape, providing a robust foundation for evaluating AI systems on scientific discovery and research comprehension.

• **Step 2: Input Construction for AI-Researcher**. To generate input for AI Agent systems, we emulate the scholarly research approach of first reviewing literature extensively before formulating new directions. We construct input information from two complementary dimensions: i) **Reference Literature Review**: domain-specific references providing knowledge foundation; and ii) **Research Requirements**: strategic objectives that direct the Agent toward targeted scientific discovery paths.

**Reference Literature Review**. Understanding the scientific research process is essential for developing effective AI research systems. Just as human researchers begin by exploring relevant literature before conducting their own investigations, our AI-Researcher model follow a similar path. Identifying key influences on scientific advancement requires rigorous methodology. Our process aims to distill the 15-20 references $\mathcal{R}$ that fundamentally influenced each target paper $y$, revealing the intellectual foundations upon which breakthroughs are built. By extracting these critical references, we construct appropriate inputs for our AI-Researcher that mirror the human research process.

We prioritize references that provide methodological frameworks, contribute essential components, or inspire conceptual innovations—elements that illuminate the paper's scientific lineage. To ensure both systematic rigor and objective assessment of reference importance, we have implemented a comprehensive five-step LLM-based evaluation process for reference input selection as follows: i) **Citation pattern analysis**: Quantify citation frequency and section distribution to identify strategically placed references. ii) **Context analysis**: Evaluate each reference's influence on methodology, theory, and experimental design. iii) **Evidence collection**: Gather specific textual evidence demonstrating reference impact for transparent verification. iv) **Impact scoring**: Compute importance scores through integrated analysis of quantitative and qualitative factors. v) **Final selection**: Select and justify the top 15-20 references that demonstrably shaped the paper's contributions.

In detail, this reference extraction is a multi-step procedure comprising five distinct steps, accompanied by a overall task description. The corresponding prompt is presented below.

**Step 1:**

Listing 20: Prompt of Step 1 in **Reference Extraction**

```
[STEP 1: Citation Pattern Analysis]
Create a statistical map of citations in the paper:
- Count citation frequency
- Track citation locations (which sections)
- Note cross-section citations
- List at least 15 most frequently cited papers

Output format:
{
    "citation_map": [
        {
            "reference": "the exact paper title in references",
            "count": number,
            "sections": ["section names"],
            "quotes": ["citation contexts"]
        }
    ]
}
```

**Step 2:**

Listing 21: Prompt of Step 2 in **Reference Extraction**

```
[STEP 2: Context Analysis]
Analyze how each frequently cited paper is discussed:
- Look for influence indicators (e.g., "based on", "extends")
- Assess discussion depth
- Identify methodology-related citations

Output format:
{
    "context_analysis": [
        {
            "reference": "the exact paper title in references",
            "indicators": ["relevant phrases"],
```

```
            "depth": "detailed/moderate/brief",
            "is_method": boolean,
            "quotes": ["key contexts"]
        }
    ]
}
```

**Step 3:**

Listing 22: Prompt of Step 3 in **Reference Extraction**

```
[STEP 3: Evidence Collection]
For each significant citation, identify:
- Concepts/methods borrowed
- How they were modified/improved
- Evidence of influence

Output format:
{
    "evidence": [
        {
            "reference": "the exact paper title in references",
            "borrowed": ["elements used"],
            "changes": ["modifications made"],
            "evidence": ["supporting quotes"],
            "type": "foundation/component/inspiration"
        }
    ]
}
```

**Step 4:**

Listing 23: Prompt of Step 4 in **Reference Extraction**

```
[STEP 4: Impact Scoring]
Score each reference based on:
- Citation frequency (30%)
- Location importance (25%)
- Discussion depth (25%)
- Direct influence (20%)

Output format:
{
    "scores": [
        {
            "reference": "the exact paper title in references",
            "total": number,
            "breakdown": {
                "frequency": number,
                "location": number,
                "depth": number,
                "influence": number
            }
        }
    ]
}
```

**Step 5:**

Listing 24: Prompt of Step 5 in **Reference Extraction**

```
[STEP 5: Final Selection]
Select and justify top 15-25 most influential papers:
- Rank based on impact scores
- Provide detailed justification
- Include specific evidence
- Explain critical importance
```

```
Output format:
{
    "top_papers": [
        {
            "reference": "the exact paper title in references",
            "rank": number,
            "type": ["methodological/component/conceptual"],
            "justification": "detailed explanation",
            "usage": "how paper was used"
        }
    ]
}
```

**Overall:**

Listing 25: Prompt of overall task description in **Reference Extraction**

```
[OVERALL INSTRUCTION]
Identify the most influential references in this research paper based on three
↪ criteria:
1. Methodological Foundation - Papers that provided core methods
2. Critical Components - Papers whose specific techniques were integrated
3. Conceptual Inspiration - Papers that shaped the research direction
```

**Research Requirement Generation**. To formulate the research directive $I$, we employ LLMs to extract the fundamental research concept from the target paper $y$. This systematic extraction identifies the core research focus, existing limitations, critical challenges, and primary objectives–effectively capturing the study's essential contributions and underlying motivations. To maintain scientific integrity and prevent information leakage, we carefully exclude all technical specifications, model identifiers, quantitative results, and architectural details from the research directive.

Specifically, the following prompt is used to generate a level 1 input idea:

Listing 26: Prompt to extract the detailed idea of a given target paper.

```
Analyze the given research paper and write a detailed technical instruction
↪ paragraph for researchers to implement its core methodology without reading
↪ the full paper. Your instruction must include:

1. What task does the model work on
2. Core techniques/algorithms used in the paper (e.g., specific neural network
↪ architectures, optimization methods, data processing approaches)
3. Purpose and function of each major technical component
4. Implementation details for each component, such as:
   - Key parameters and configurations
   - Input/output specifications
   - Important constraints or requirements
5. Step-by-step description of how these components interact and combine into
↪ the complete system
6. Critical implementation details that affect performance

(If the examples above do not apply to the input paper, ignore the examples)

Focus only on the technical methodology and implementation aspects. Exclude
↪ background information, literature review, and experimental results. Write
↪ in a clear, sequential format that a technical researcher could follow to
↪ reproduce the core method.

Directly write the instruction without any other words.

Don't mention the specific names of the proposed model, or exact module names
↪ that are special to this paper.
```

- **Step 3: Rigorous Anonymization to Ensure Scientific Originality**. A critical challenge in evaluating AI research agents lies in distinguishing between genuine problem-solving abilities and mere regurgitation of memorized content. To address this fundamental concern, we implement a comprehensive anonymization protocol that transforms the evaluation landscape: i) **Method name masking**: Replace algorithm and model names with generic identifiers, testing conceptual understanding rather than term recognition. ii) **Technical detail abstraction**: Remove implementation specifics while preserving core concepts, requiring engagement with fundamental principles. iii) **Dataset standardization**: Normalize experimental contexts to create a fair evaluation landscape that prevents shortcuts based on dataset familiarity. iv) **Citation anonymization**: Eliminate temporal and institutional markers to test problem-solving rather than information recall.

### A.7.3 Evaluation of AI-conducted Scientific Discovery

To rigorously assess the genuine scientific discovery capabilities of AI agent system on our Scientist-Bench benchmark, we implement a two-stage evaluation framework that addresses both technical implementation fidelity and scientific innovation merit.

- **Stage 1: Technical Execution Validation**. The first stage employs a specialized code review agent to verify whether the implementation code $\mathcal{C}$ faithfully realizes the AI-conducted research innovations. This critical verification prevents scenarios where AI researchers might propose sophisticated methods with promising results without providing functional implementations—a fundamental requirement for credible scientific discovery. The code review agent performs static analysis and runtime verification across key dimensions including *Algorithm Correctness*, *Computational Efficiency*, and *Adherence to Specified Constraints*. We quantify this assessment using a completion ratio metric that reflects the proportion of required functionality successfully implemented by the AI researcher.

- **Stage 2: Scientific Contribution Evaluation.** The second stage rigorously assesses whether AI agent systems have produced genuine scientific innovations by comparing the generated research report $p$ against the groundtruth target paper $y$. To ensure objective evaluation of scientific merit, we implement a structured comparison protocol:

$$r, J = \text{PaperReview}(\text{RandomSwap}(p, y); g) \tag{1}$$

This formulation employs a calibrated paper review agent that produces a comparative rating $r \in \{-3, -2, -1, 0, 1, 2, 3\}$, where positive values indicate the AI-generated paper exceeds the target paper in scientific contribution, zero indicates equivalence, and negative values signal inferior quality. The magnitude of $r$ quantifies the degree of scientific advancement or regression. The review agent also provides $J$, a structured set of justifications based on reviewing guidelines $g$ derived from ICLR conference standards—widely recognized in the machine learning community for emphasizing originality, technical soundness, and significance of contributions.

To ensure methodological rigor, we incorporate two critical debiasing mechanisms: (1) randomly swapping the presentation order of papers to eliminate position bias, and (2) conducting multiple independent evaluations using diverse state-of-the-art LLMs (including multiple GPT, Claude, and Gemini models) with temperature set to 1, creating a comprehensive panel-like review process that effectively mitigates individual model biases and enhances evaluation reliability. This carefully designed approach establishes a robust framework for quantifying whether AI systems can independently discover scientific insights that match or exceed those produced by human researchers. The corresponding prompts are presented below.

Listing 27: Prompt to extract the model's name of a given target paper.

```
Given a research paper's title and abstract, extract the name of the novel
↪ model/method introduced in the paper:

1. Look for phrases that signal a new model introduction, such as:
   - "we␣propose/present/introduce"
   - "our␣model/method/approach"
   - "called/named"
   - Model name followed by model architecture details

2. Return format:
   - If a proposed model name is found: Return only the model name
```

```
    - If you find both abbreviation and full name for the model, format them into
    ↪ "abbreviation,␣full␣name"
    - If no clear model name is found: Return "NO␣MODEL␣NAME␣FOUND"
    - You should strictly follow the requirement, and output without any other
    ↪ words

3. Focus only on the main proposed model:
    - Ignore baseline models
    - Ignore models from referenced papers
    - Ignore general model categories/types

Input:
- Paper Title: {paper_title}
- Paper Content: {paper_content}
```

Listing 28: Remove the model name of the input instruction.

```
Given a research paper's proposed model name and its paper title, anonymize any
↪ mentions of the model name and direct paper self-references in a paragraph.

Replace:
- Model name and variations with "the␣proposed␣model" or "the␣proposed␣approach"
- Paper self-references with "this␣paper" or "this␣study"
- Keep all other content exactly as written

Input:
- Paper Title: {paper_title}
- Model Name: {model_name}
- Paragraph: {paragraph}

Output:
- If no model name mentions found: Return "NO␣NEED␣TO␣PROCESS"
- If anonymization needed: Return the processed paragraph with only required
↪ replacements

Note: Only anonymize the specified model name and direct paper references. Keep
↪ all other content, including other model names and references, unchanged.
```

**Alignment with Top-Tier Peer-Review Standards**. To ensure our evaluation framework upholds rigorous academic standards, we align Scientific Contribution Evaluation with established peer-review protocols from premier venues. Specifically, we assess research quality across critical dimensions, including technical novelty, methodological rigor, empirical validation, and impact–directly mirroring comprehensive evaluation criteria used in the ICLR conference review process.

To validate the reliability of our LLM-based evaluation mechanism, we conducted extensive benchmark experiments on a diverse sample of previously published ICLR papers with known acceptance decisions, demonstrating strong correlation between our automated assessments and the judgments rendered by expert human reviewers in real-world academic settings. Experiments using 5 popular LLMs with 64 randomly sampled ICLR submissions—forming 32 paper pairs for comparison—demonstrate that our LLM-based reviewer judgments perfectly align with ICLR's final decisions, in pairwise paper reviewing and confirming robust alignment with top-tier peer-review standards.

## A.8 Experimental Settings

**Experimental Dataset: Benchmarking Scientific Innovation**. We evaluate our AI-Researcher using the Scientist-Bench benchmark (as described in Section 2)–a meticulously curated collection of 22 state-of-the-art papers spanning several critical AI domains including Computer Vision (*e.g.*, Diffusion Model), Signal Processing (*e.g.*, Vector Quantization), Graph Learning (*e.g.*, Graph Neural Networks), and Information Retrieval (*e.g.*, Recommender Systems). Our work addresses a significant gap in the field, as comprehensive benchmarks for scientific innovation assessment remain notably scarce. The evaluation protocol employs **Two Complementary Innovation Tasks of Varying Difficulty Levels** (detailed below) designed to test distinct research capabilities across diverse

Table 5: Data statistics of Scientist-Bench across diverse research domains, featuring comprehensive task distribution across guided innovation and open-ended exploration challenges.

| Research Domain | # Papers | # Level-1 | # Level-2 | # Rejected Papers |
|---|---|---|---|---|
| Diffusion Models | 4 | 4 | 1 | 0 |
| Vector Quantization | 6 | 6 | 1 | 0 |
| Graph Neural Networks | 7 | 7 | 1 | 1 |
| Recommender Systems | 5 | 5 | 3 | 1 |
| Total | 22 | 22 | 6 | 2 |

methodological paradigms. Table 5 presents the complete dataset statistics, establishing important baseline measures in this underexplored evaluation landscape for future comparative analyses.

• **Level-1 Task: Guided Innovation** — The scientific discovery agent receives explicit research instructions alongside reference papers, simulating scenarios where researchers pursue specific innovation targets. This structured evaluation provides clear assessment of the agent's ability to develop targeted innovations and spans all 22 groundtruth papers for comprehensive coverage.

• **Level-2 Task: Autonomous Exploration** — The scientific discovery agent performs independent, open-ended research exploration with only reference papers as input. This more challenging scenario tests the agent's capacity to identify promising research gaps and generate novel directions without explicit guidance—a crucial capability for truly autonomous scientific assistants. To maintain methodological integrity and prevent cross-contamination from overlapping reference materials, we strategically selected 5 groundtruth papers across distinct research domains, enabling us to rigorously assess genuine discovery abilities without confounding influences.

**Evaluation Protocols**. To assess scientific contributions of AI-Researcher, we implement a two-stage evaluation framework examining both technical implementation and research quality:

• **Implementation Verification**. We employ a specialized code review agent to verify that AI-generated code faithfully implements the proposed methodology described in the technical report. This critical validation step ensures the practical reproducibility of the research contribution. We quantify performance using the completion ratio $R$, defined as the fraction of AI implementations correctly executing the intended research approach. This metric directly measures the model's ability to translate conceptual innovations into functional implementations.

• **Scientific Quality Assessment**. For implementations that successfully pass verification, we perform an in-depth comparative analysis between AI-generated research and their human-authored counterparts. This evaluation mirrors the rigorous peer-review process typical at prestigious venues like ICLR and NeurIPS Jin et al. [2024], where an expert review agent systematically examines each paper pair through three fundamental scientific dimensions: • (1) innovation and novelty of research contributions, • (2) theoretical and methodological rigor, and • (3) empirical validation and experimental design quality. This approach ensures our assessment adheres to established standards of scientific excellence in the field. The evaluation culminates in a comprehensive comparative rating on a 7-point scale (-3 to +3), where negative scores indicate AI work falling below human standards, zero represents parity, and positive scores signify AI research exceeding human benchmarks. Each rating is substantiated with detailed justifications citing specific evidence from both papers, providing transparent rationale for the comparative assessment.

**LLMs as Judges with Robust Evaluation**. To establish robust evaluation, we leverage five state-of-the-art LLMs (GPT-4, o1-mini, o3-mini, Claude-sonnet-3.5, and Claude-sonnet-3.7), each performing 16 independent assessments per paper with temperature=1.0. This ensemble approach mitigates individual model biases and provides statistical confidence in our findings. We analyze results through two complementary metrics: (1) mean rating across all evaluations—quantifying the quality gap between AI and human research, and (2) percentage of AI papers scoring $\geq$-1.0—representing research contributions that achieve at least near-human quality in the field.

### A.9 Paper Review Agent Validation Against Human Expert Judgments (RQ5)

To rigorously validate our automated review system's alignment with expert scientific assessment, we conducted a systematic evaluation using gold-standard human judgment data from the ICLR

Table 6: Paper Review Agent Alignment with Human Expert Decisions.

| Year | Metric | Gemini-2.0-flash | GPT-4o | o3-mini | Claude-3.5 | Claude-3.7 |
|---|---|---|---|---|---|---|
| 2021 | Average Rating | 0.33±1.51 | 0.12±0.95 | 0.64±0.89 | 0.73±1.11 | 0.66±1.68 |
| | Comparable(%) | 100.00% | 100.00% | 100.00% | 100.00% | 100.00% |
| | Acc Better(%) | 71.43% | 71.43% | 85.71% | 85.71% | 85.71% |
| 2022 | Average Rating | 0.38±1.65 | 0.41±0.89 | 0.79±0.88 | 1.20±0.90 | 0.64±1.42 |
| | Comparable(%) | 100.00% | 100.00% | 100.00% | 100.00% | 100.00% |
| | Acc Better(%) | 60.00% | 90.00% | 90.00% | 90.00% | 80.00% |
| 2023 | Average Rating | 0.25±1.71 | 0.33±0.97 | 0.67±0.85 | 0.97±1.11 | 0.73±1.48 |
| | Comparable(%) | 86.67% | 100.00% | 100.00% | 100.00% | 100.00% |
| | Acc Better(%) | 66.67% | 80.00% | 93.33% | 93.33% | 80.00% |
| Overall | Average Rating | 0.31±1.65 | 0.31±0.95 | 0.70±0.87 | 0.99±1.06 | 0.69±1.51 |
| | Comparable(%) | 93.75% | 100.00% | 100.00% | 100.00% | 100.00% |
| | Acc Better(%) | 65.62% | 81.25% | 90.62% | 90.62% | 81.25% |

conference. We constructed a validation dataset comprising 32 carefully sampled paper pairs from proceedings (2021-2023), where each pair contains one accepted and one rejected submission. To ensure meaningful comparative analysis and maintain consistency with our main experimental protocol, we prioritized paper pairs exhibiting high TF-IDF similarity in content and focus.

We applied identical pairwise review methodology as our main experiments, evaluating performance through three complementary metrics: (1) discriminative rating (scale of -3 to 3, with positive values indicating higher ratings for accepted papers), (2) comparable quality detection (percentage of pairs rated above -1.0), and (3) acceptance prediction accuracy (percentage of pairs where accepted papers received ratings above 0.0). Table 6 presents the comprehensive validation results, revealing several key insights into our review agent's judgment capabilities:

● **Robust Expert-Aligned Evaluation Capabilities**. Our paper review agent demonstrates consistent discriminative validity across independent evaluations, with all evaluator models producing positive mean ratings (0.31-0.99) when comparing accepted versus rejected papers. This consistent directional alignment validates the system's fundamental quality assessment capabilities. The evaluators achieve near-perfect comparable rate identification (100% for all models except Gemini-2.0-flash), confirming the agent's reliability in recognizing legitimate scholarly contributions even in rejected papers.

Most significantly, the system demonstrates strong decision alignment with expert conference reviewers, correctly identifying the superior paper in 65.62% to 90.62% of cases, with five of six evaluators exceeding 81% accuracy across the 32 paper pairs. This exceptional concordance with human expert decisions provides compelling evidence that our automated review agent captures the nuanced quality distinctions that drive scientific peer review decisions.

● **Differential Reliability Across Evaluation Models**. Systematic performance analysis reveals substantial variation in evaluator alignment with human expert judgments. Gemini-2.0-flash demonstrates notably inferior reliability metrics—exhibiting both the lowest average rating and highest standard deviation among all tested models—which necessitated its exclusion from our primary experimental evaluations. In contrast, all other LLM evaluators achieved perfect comparable rate identification (100%), providing strong methodological justification for their inclusion in our AI-generated research assessment protocol. Particularly noteworthy is the comparative performance between Claude-3.5 and Claude-3.7, where the latter's enhanced system-2 thinking capabilities did not translate to superior review performance, suggesting that deliberative reasoning features may not significantly benefit scientific quality assessment tasks compared to other model capabilities.

### A.10   Case Studies of AI-Generated Scientific Contributions

We focused our analysis on the `rotation_vq` benchmark task, using our standard configuration of Claude-series models for experimentation and implementation paired with the 4o model for manuscript generation. This detailed examination of actual system outputs reveals several noteworthy characteristics about the nature and quality of AI-conducted research:

● **Structured Software Architecture with Minimal Scaffolding**. Figure 6 illustrates the remarkably organized project architecture produced by our `Code Agent`, featuring systematically decoupled

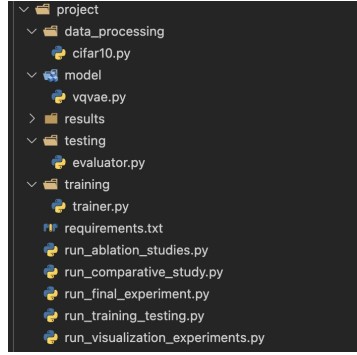

Figure 6: [a] Code Structure.

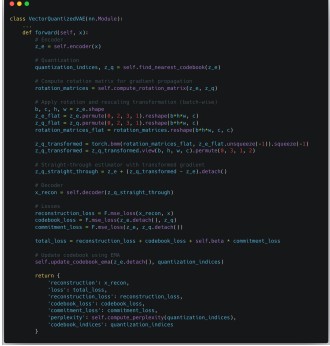

Figure 7: [b] Code Samples

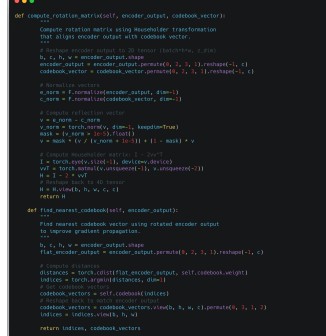

Figure 8: [c] Code Samples

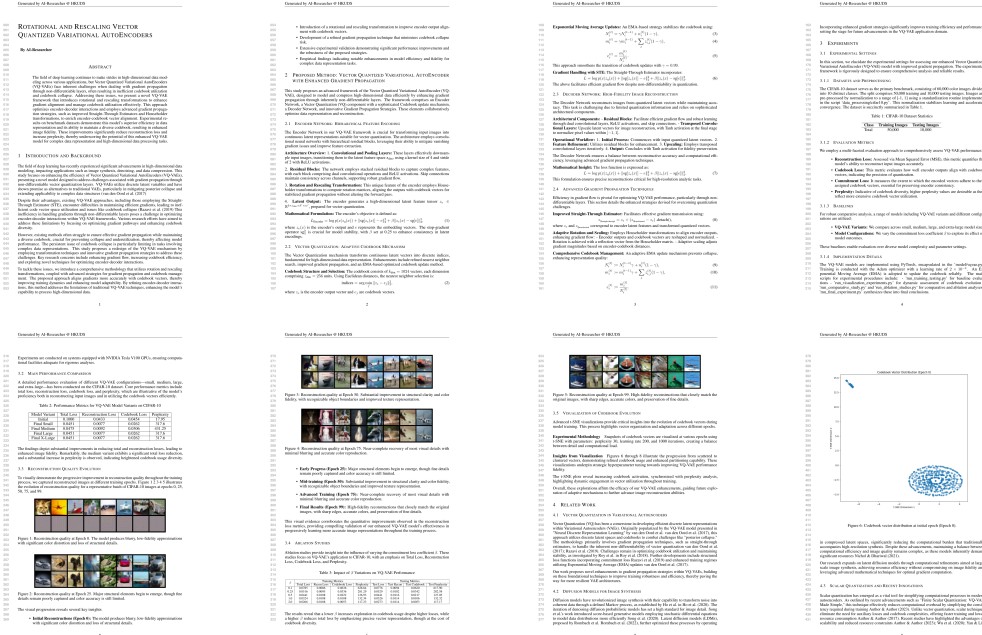

Figure 9: Case Studies of AI-Generated Scientific Contributions by AI-Researcher.

model components, training pipelines, and evaluation modules with well-defined entry points. This architectural clarity stems from our balanced approach to agent guidance—providing high-level structural suggestions rather than rigid templates. Unlike previous frameworks that require agents to adapt pre-existing codebases, our methodology allows agents to develop implementations from first principles while incorporating best practices from human software engineering. This approach demonstrably enhances implementation coherence while minimizing the cognitive overhead associated with codebase familiarization. As evidenced in Figures 7 and 8, the resulting implementations exhibit professional coding standards with comprehensive documentation and logical modularization that facilitates both reproducibility and extensibility.

• **Emergent Experimental Thoroughness Without Explicit Directives**. A notable capability of our AI-Researcher framework is its autonomous experimental design process that emerges through collaborative interactions between the `Code Agent` and `Advisor Agent`. Rather than prescribing a predetermined protocol, our system progressively develops comprehensive evaluation strategies through iterative refinement. This self-directed experimental thoroughness is evident in the final manuscript shown in Figure 9, where the system independently conducts and reports a complete scientific evaluation including overall performance benchmarking, controlled ablation studies, training dynamics visualization, and latent space embedding analysis. This comprehensive experimen-

tal methodology emerges organically from the multi-agent system without explicit experimental requirements–demonstrating sophisticated scientific reasoning beyond simple instruction following.

## A.11   Related Work

### A.11.1   AI Agent Systems

AI agent frameworks have evolved through three distinct architectural paradigms.

• **Tool Integration Frameworks**. The first paradigm establishes foundational integration layers for AI components. LangChain Contributors [2023] introduced standardized interfaces enabling seamless interoperability between models, embeddings, and vector stores within workflows. HuggingGPT Shen et al. [2023] leveraged this approach by positioning LLMs as orchestration controllers that coordinate specialized models from Hugging Face ecosystem. OpenAgents Xie et al. [2023] democratized these capabilities by providing domain-specific agents for data analysis, API integration, and web browsing for non-expert users.

• **Multi-Agent Collaboration Frameworks**. The second paradigm addresses complex problem solving through structured agent interactions. MetaGPT Hong et al. [2024] formalized human work-flow patterns through Standardized Operating Procedures (SOPs), creating systematic collaboration protocols. AutoGen Microsoft AutoGen Team [2025] expanded this vision with a comprehensive programming framework for developing systems that support both autonomous operation and human collaboration. AgentScope Gao et al. [2024] prioritized robust coordination through a message-exchange architecture with built-in fault tolerance. CAMEL introduced innovative role-playing techniques that facilitate autonomous agent cooperation while maintaining alignment with human intentions.

• **Self-Directed Agentic Task Execution Systems**. The third paradigm focuses on agents capable of independent goal pursuit with minimal supervision. Agentic AI systems like Manus Manus Technologies [2025] and open-source alternatives including OpenManus OpenManus Contributors [2025] and OWL Li et al. [2023] extend these capabilities to handle complex online tasks without continuous human intervention. AutoAgent Tang et al. [2025] represents the frontier–a fully-automated, zero-code approach functioning as an Agent Operating System enabling non-technical users to create agents using natural language alone.

Agent frameworks have evolved from isolated systems to sophisticated multi-agent architectures with specialized coordination. However, **these systems fundamentally lack the intellectual capacity for true scientific innovation**. Despite advances, they remain insufficient for scientific discovery because such work requires **a level of intelligence that transcends current capabilities**. Scientific breakthroughs demand **nuanced hypothesis formation, creative experimental design, understanding and implementation of complex algorithms, and critical synthesis of knowledge**–cognitive processes requiring deeper reasoning and domain expertise than existing systems can provide.

### A.11.2   AI-Driven Research Systems

Recent advances have transformed AI's role in scientific research from assistive tools to autonomous agents capable of executing complete research workflows. The AI Scientist framework Lu et al. [2024] pioneered this field as the first comprehensive system where frontier language models independently generate research ideas, conduct experiments, and produce scientific papers. Complementary approaches include CycleResearcher Weng et al. [2025], which demonstrated the viability of open-source LLMs for autonomous research through a complete cycle from literature review to refinement, and the AI co-scientist AI Co-Scientist Team [2025], which employs multi-agent debate and evolution mechanics to generate novel scientific hypotheses with promising applications in biomedical domains.

These systems are supported by emerging research platforms and evaluation frameworks that enhance their capabilities and measure their effectiveness. Agent Laboratory Schmidgall et al. [2025] provides an end-to-end autonomous research workflow with specialized LLM agents assisting humans through literature review, experimentation, and report writing, while its extension AgentRxiv Schmidgall and Moor [2025] enables collaborative scientific progress by allowing agents to share and build upon each other's work. Collectively, these developments represent a paradigm shift toward automated scientific inquiry, though matching human-level research capabilities remains an ongoing challenge that requires further advancement.

### A.12 Limitations and Future Directions

While AI-Researcher demonstrates strong capabilities in automating the end-to-end scientific research process, several important limitations remain that warrant future investigation:

**Ethical and Safety Considerations in Autonomous Research.** Our current framework primarily focuses on technical feasibility and performance metrics, without a systematic treatment of the ethical, societal, and safety implications of fully autonomous scientific discovery. As AI-generated hypotheses, implementations, and publications become more prevalent, it is critical to address potential concerns such as biased scientific reasoning, unsafe experimental suggestions, and the accountability of non-human authorship. Future work should integrate robust ethical guardrails, human-in-the-loop oversight mechanisms, and value alignment strategies to ensure responsible deployment of autonomous research systems.

**Limited Diversity in LLM Backbones.** The present evaluation of AI-Researcher primarily relies on a single class of LLM backbone, which may constrain the generalizability of the observed findings. Different foundation models may exhibit varied strengths in reasoning, coding, and scientific communication. Expanding the system to support and benchmark across a broader spectrum of LLMs—including open-source and multilingual variants—could provide deeper insights into model-specific capabilities and enhance the robustness and adaptability of autonomous research agents.

