# OpenReview forum: "AI-Researcher: Autonomous Scientific Innovation"
_NeurIPS.cc/2025/Conference — NeurIPS 2025 spotlight_

### Official Review · Reviewer_875E · 2025-06-22

**Clarity:** 3
**Significance:** 3
**Originality:** 3
**Rating:** 4
**Confidence:** 4

**Summary:**

This paper proposes an autonomous research system, named AI-Researcher, with seamless research pipeline: literature review, hypothesis generation, algorithm implementation and publication-ready manuscript. To evaluate the performance of the framework, the paper also introduces a benchmark, Scientist-Bench, comprising state-of-the-art papers across diverse AI research domains.

**Questions:**

Additional Feedback

1.	To improve clarity, the figure should explicitly depict the internal components of each agent, along with the flow of information between them. Furthermore, vague terms such as “taking notes” lack definition—please clarify their meaning and role within the research pipeline.
2.	Certain elements in the paper lack proper citation. For instance, Table 1 is not cited or discussed in the main text, which reduces clarity and coherence.
3.	The writing should be improved more to enhance readability.

**Ethical Concerns:**

["NO or VERY MINOR ethics concerns only"]

**Final Justification:**

The rebuttal has addressed my concerns, and I am inclined to increase my score. However, due to the recent reminder to reviewers, “Any rebuttals that were posted in their entirety as comments after Jul 30 are to be ignored,” I will need to disregard your earlier comments. Therefore, I am keeping my original borderline accept rating. The authors may check with the AC to confirm whether the score can be increased in this case.

**Limitations:**

Yes

**Quality:**

3

**Strengths And Weaknesses:**

Strengths and Contributions

1.	The topic of autonomous research is meaningful and interesting.
2.	The paper also introduces a new benchmarking to evaluate the performance of autonomous AI research.

Limitations / Weaknesses

1.	Some important discussions are missing.
Autonomous scientific discovery is not a new topic, multiple existing methods (with whole research pipeline) are available, as illustrated in your appendix A.11.2. Discussion and highlighting the novelty of this paper with existing methods are necessary.

2.	Missing comparisons with existing AI scientist frameworks.
The paper does not include direct comparisons with other AI scientist systems (e.g., [1], [2]), which limits the ability to assess relative performance and novelty.

3.	Missing baseline comparisons with other foundation models.
While the paper reports performance differences between GPT and Claude series, it lacks a broader baseline comparison. For a more comprehensive evaluation, results from additional foundation models—such as Gemini, or DeepSeek—should be included to validate the generality and robustness of the proposed framework across architectures.

4.	Missing experimental details and unclear evaluation criteria.
In line 304, the paper states that it evaluates aspects such as research motivation, methodology, innovation, and experimental validation. However, it is unclear how these dimensions are assessed—whether through human annotation, LLM-based scoring, or other means. Additionally, the interpretation of the scores presented in the corresponding table is not explained. Please clarify the evaluation protocol and the meaning of the numerical results.

5.	Applicability to broader scientific domains is unclear.
The current work focuses primarily on AI-related tasks. It remains uncertain whether the proposed framework generalizes to other scientific fields such as biomedicine, chemistry, or physics. Please clarify the system’s domain adaptability and any assumptions or limitations that may restrict its applicability.


[1]. Lu, C., Lu, C., Lange, R. T., Foerster, J., Clune, J., & Ha, D. (2024). The ai scientist: Towards fully automated open-ended scientific discovery. arXiv preprint arXiv:2408.06292.

[2]. Weng, Y., Zhu, M., Bao, G., Zhang, H., Wang, J., Zhang, Y., & Yang, L. (2024). Cycleresearcher: Improving automated research via automated review. arXiv preprint arXiv:2411.00816.

---

> ### Author Response · Authors · 2025-08-02
> **Response to Reviewer 875E (1/2)**
>
> # Response to Reviewer 875E
>
> ## Discussion of Novelty Compared to Existing Methods
>
> While autonomous scientific discovery has been explored by several existing frameworks as detailed in Section A.11.2, our AI-Researcher introduces several key innovations that distinguish it from prior work. Unlike existing systems that focus on specific components of the research pipeline, our framework provides the first truly end-to-end automated scientific research system capable of generating publication-ready papers from initial concepts without human intervention. While AI Scientist pioneered automated research idea generation and experimentation, and CycleResearcher demonstrated open-source LLM viability, these systems lack the comprehensive integration and systematic evaluation framework that our work provides.
>
> Our key contributions include: (1) A complete end-to-end automation pipeline that produces publication-quality manuscripts, surpassing the partial automation offered by existing frameworks; (2) Scientist-Bench, a systematic evaluation framework containing 22 research papers across multiple domains, enabling standardized assessment of autonomous research capabilities at both guided (Level-1) and open-ended (Level-2) task levels; (3) An innovative Code-Advisor Agent iterative optimization paradigm that significantly improves algorithm implementation accuracy through structured feedback cycles, addressing a critical limitation in existing autonomous research systems. These innovations collectively enable more reliable and comprehensive autonomous scientific discovery compared to current approaches.
>
> ## Comparison with AI Scientist Frameworks and Different LLM Backbones
>
> | Variants                        | Correctness |
> | ------------------------------- | ----------- |
> | AI-Researcher + Claude-series   | 3.22        |
> | AI-Researcher + Gemini-series   | 3.06        |
> | AI-Researcher + 4o-series       | 2           |
> | AI-Researcher + Deepseek-series | 2.39        |
> | AI-Scientist + Claude-series    | 2.89        |
>
> Thank you for pointing out these important missing comparisons. To address both concerns, we conducted additional experiments comparing our AI-Researcher system against existing frameworks and with different LLM backbones.
>
> **Different** **LLM** **Backbones:** We evaluated our framework using different LLM series on Level-1 tasks in the vector quantization domain. As shown in Table 1, our system achieves best performance with Claude-series (`claude-3-5-sonnet-20241022` + `claude-3-5-haiku-20241022`, 3.22), followed by Gemini-series (`gemini-2.5-pro` + `gemini-2.5-flash`, 3.06), DeepSeek-series (`deepseek-r1-0528` + `deepseek-v3-0324`, 2.39), and 4o-series (`gpt-4o-2024-08-06` + `gpt-4o-mini-2024-07-18`, 2.0). Gemini-series occasionally encounters agent loop stagnation issues, while DeepSeek-series and 4o-series suffer from implementation hallucinations where agents incorrectly believe they have successfully implemented solutions.
>
> **AI** **Scientist Framework Comparison:** Our AI-Researcher system (using Claude-series) significantly outperforms the AI-Scientist framework (using Claude-series) (2.89 correctness score) on the same tasks. This superior performance stems from our explicit code-advisor agent paradigm, which enables reflection and iterative correction in each loop, substantially improving implementation accuracy compared to systems without structured feedback mechanisms.
>
> These results demonstrate both the robustness of our framework across different foundation models and its superiority over existing autonomous research systems.
>
> ## Evaluation Protocol and Score Interpretation
>
> The evaluation dimensions (research motivation, methodology, innovation, and experimental validation) are embedded within our ICLR-based review guidelines that guide the LLM review agents' assessment process. The evaluation is conducted entirely through LLM-based scoring, where review agents perform pairwise comparisons between AI-generated and human papers using these guidelines. The numerical scores represent comparative ratings on a 7-point scale (-3 to +3), where negative values indicate AI papers are inferior to human papers, zero indicates equivalence, and positive values indicate superiority. Each score reflects the aggregated assessment across all four evaluation dimensions rather than separate scoring per dimension.

---

> > ### Author Response · Authors · 2025-08-02
> > **Response to Reviewer 875E (2/2)**
> >
> > ## Domain Adaptability and Limitations
> >
> > Thank you for raising this important concern about our framework's applicability to broader scientific domains beyond AI research. We acknowledge that our current evaluation focuses primarily on AI-related tasks, which may raise questions about generalizability.
> >
> > However, our AI-Researcher system is designed with inherent generalizability in mind. The framework covers the fundamental scientific research pipeline—literature review, idea generation, experimental planning, and validation—which represents core processes consistent across scientific disciplines including biomedicine, chemistry, and physics. The modular multi-agent architecture facilitates domain-specific adaptations while preserving the underlying scientific methodology.
> >
> > For broader applicability, our system can handle computational and analytical tasks across different scientific domains, as the fundamental research workflow patterns translate effectively. Domain-specific adaptations would include: specialized databases (PubMed for biomedical research, chemical databases for chemistry), field-specific computational tools and libraries, and domain-appropriate evaluation metrics.
> >
> > We acknowledge key limitations: the system is currently restricted to computational research tasks and cannot perform physical laboratory experiments. Additionally, domain-specific scientific reasoning patterns and specialized mathematical formulations may require further refinement. Future work will focus on systematic evaluation across multiple scientific domains to validate and improve the framework's cross-disciplinary effectiveness.
> >
> > ## Additional Feedback Responses
> >
> > Thank you for these constructive suggestions to improve our paper's clarity and presentation. We have addressed each concern:
> >
> > - Figure and Terminology Clarity: We have enhanced figures to explicitly show internal agent components and information flow between them, while clarifying vague terms such as "taking notes" which has been revised to "save into long term memory" to better reflect the actual functionality within the research pipeline.
> > - Citation and Reference Integration: We have ensured all tables and figures are properly cited and discussed in the main text, including Table 1 and other previously uncited elements to improve coherence.
> > - Writing and Readability: We have conducted comprehensive revision to improve writing clarity, sentence structure, and logical flow throughout the manuscript while maintaining technical rigor.

---

> > > ### Comment · Reviewer_875E · 2025-08-04
> > >
> > > Thank you to the authors for their explanation. I have the following concerns:
> > >
> > > **Clarification of Novelty Compared to Existing Methods**
> > >
> > > I would appreciate further clarification on how this work differs from other automated research frameworks, such as AI-Scientist and CycleResearcher. Beyond introducing a systematic evaluation framework, what is the core methodological novelty of your approach?
> > >
> > > **Comparison with AI Scientist Frameworks and LLM Backbones**
> > >
> > > Thank you for providing the comparison results. However, I am particularly interested in how your method compares with other state-of-the-art AI scientist frameworks using your review agents (which rate from -3 to +3). A more direct comparison on this front would be helpful.
> > >
> > > **Additional Question: Clarification on Ground-Truth Papers**
> > >
> > > Could you elaborate on how the ground-truth papers were selected for evaluation? Specifically, what are their publication dates? Is there a risk that the LLMs used may have seen these papers during training?
> > >
> > > **If the authors can address these concerns, I would be inclined to raise my score.**

---

> > > > ### Author Response · Authors · 2025-08-05
> > > > **Response to Reviewer 875E's Concerns (2/2)**
> > > >
> > > > ## Ground-Truth Paper Selection and Knowledge Cutoff Concerns
> > > >
> > > > Thank you for this important clarification question about our ground-truth paper selection methodology and potential knowledge contamination.
> > > >
> > > > Our detailed selection logic is outlined in Appendix A.7.2 Benchmark Construction, Step 1: Systematic Selection of AI Research Benchmark Papers. Specifically, we selected 22 top-cited papers from late 2022 to present, with the majority of papers published after April 2024—which is after the knowledge cutoff of Claude 3.5 Sonnet (April 2024). This temporal selection strategy significantly reduces the risk of knowledge contamination from training data.
> > > >
> > > > To further mitigate information leakage, we implemented temporal constraints in our experimental setup: both GitHub search and paper search are restricted to information available only before each target paper's publication date, preventing access to information that could reveal the original paper's content or approach.
> > > >
> > > > More importantly, we want to clarify that our agents do not engage in needle-in-haystack retrieval to search for specific papers they may have encountered during training. Instead, they follow the autonomous research paradigm by systematically conducting literature review, generating novel ideas, and implementing solutions based on the provided reference materials and their general scientific reasoning capabilities.
> > > >
> > > > This design ensures that our evaluation measures genuine autonomous research capabilities rather than memorization or pattern matching from training data.

---

> > > > > ### Comment · Reviewer_875E · 2025-08-05
> > > > >
> > > > > The rebuttal has addressed my concerns, and I am inclined to increase my score. However, due to the recent reminder to reviewers, “Any rebuttals that were posted in their entirety as comments after Jul 30 are to be ignored,” I will need to disregard your earlier comments. Therefore, I am keeping my original borderline accept rating. The authors may check with the AC to confirm whether the score can be increased in this case.

---

> > > > > > ### Author Response · Authors · 2025-08-06
> > > > > > **Appreciation to Reviewer 875E**
> > > > > >
> > > > > > We are deeply grateful for your recognition that our rebuttal has addressed your concerns and for your thoughtful consideration of a score increase. We completely understand and respect the reviewer guidelines regarding the July 30th deadline. The additional experiments and analyses were indeed substantial undertakings that required considerable time to execute properly. While we humbly hope there might be some flexibility for the score adjustment you kindly mentioned, we sincerely appreciate your guidance and will fully respect whatever approach aligns with the review process. Your constructive feedback has been invaluable in strengthening our work.

---

> ### Author Response · Authors · 2025-08-05
> **Response to Reviewer 875E's Concerns (1/2)**
>
> # Response to Reviewer 875E's Concerns
>
> Thank you very much for your thoughtful feedback and for indicating your willingness to raise your score based on our responses. We greatly appreciate your constructive engagement and have carefully addressed each of your three specific concerns below. We believe our detailed responses demonstrate the novelty, rigor, and validity of our approach, and we hope they will support your consideration of a score revision.
>
> ## Clarification of Novelty Compared to Existing Methods
>
> Thank you for this question. We acknowledge that both AI-Scientist and CycleResearcher represent important foundational contributions to automated research. Building upon these valuable works, our core methodological novelty lies in three key areas:
>
> **1. True End-to-End Automation vs. CycleResearcher:** CycleResearcher explicitly states that "human researchers or other agents are still needed to execute the experiments" and that "all experimental results mentioned in the generated papers were fabricated (Hallucinated)." In contrast, AI-Researcher achieves complete automation with real, verifiable experimental results.
>
> **2. Bidirectional Code-Theory Refinement vs. AI-Scientist:** Our novel Code Advisor Agent Loop introduces recursive refinement between theoretical concepts and implementations through bidirectional mappings. Unlike AI-Scientist's linear pipeline, our system continuously validates mathematical formulations against code implementations, creating a feedback-driven optimization process that dramatically reduces implementation errors.
>
> **3. Atomic Resource Decomposition:** Our Resource Analyst agents decompose complex research concepts into atomic components with explicit traceability, enabling modular verification and significantly reducing hallucination risks compared to existing monolithic approaches.
>
> These innovations collectively enable the first system to conduct complete, verifiable scientific research with human-level rigor while maintaining full automation throughout the discovery lifecycle.
>
> ## Comparison with AI Scientist Frameworks Using Standardized Review Agents
>
> Thank you for requesting this more direct comparison using our review agent evaluation framework. We conducted additional experiments comparing our AI-Researcher system with the AI-Scientist framework using GPT-4o as the judge agent, applying the same Evaluation Protocols in section A.8 Experimental Settings.
>
> |                    | **AI-Researcher** | **AI-Scientist** |
> | ------------------ | ----------------- | ---------------- |
> | **Average Rating** | -0.55±1.00        | -0.92±1.70   |
> | **Comparable(%)**  | 83.33%            | 61.46%           |
>
> As shown in Table, our AI-Researcher system demonstrates superior performance across both evaluation metrics:
>
> • Average Rating: AI-Researcher achieves -0.55±1.00 compared to AI-Scientist's -0.92±1.70, indicating significantly better paper quality with lower variance
>
> • Comparable Rate: 83.33% of AI-Researcher papers achieve comparable quality to human papers, substantially outperforming AI-Scientist's 61.46%
>
> This direct comparison validates our earlier analysis of key innovations. The superior performance stems from our comprehensive design choices: (1) The Code-Advisor Agent iterative optimization paradigm provides structured feedback cycles that significantly improve implementation accuracy—a critical weakness in existing systems; (2) Our end-to-end integration enables better coherence between idea generation, implementation, and documentation phases; (3) The systematic evaluation framework (Scientist-Bench) ensures more reliable assessment of autonomous research capabilities.
>
> These results demonstrate that our innovations translate into measurable improvements in research output quality, with both higher average ratings and greater consistency in producing human-comparable research papers compared to existing state-of-the-art autonomous research frameworks.

---

### Official Review · Reviewer_6z2Z · 2025-07-03

**Clarity:** 4
**Significance:** 3
**Originality:** 3
**Rating:** 5
**Confidence:** 4

**Summary:**

The authors build a prompt-only, multi-agent LLM baseline that -- given a capsule of 15-20 reference papers (and optionally an instruction
-- (i) writes runnable code and
-- (ii) drafts a full LaTeX manuscript.
They wrap this in Scientist-Bench, a benchmark that evaluates such pipelines through two sequential gates:

Implementation gate -- Did the code compile & run? Was the output logically consistent (1-to-5 “Correctness” from an LLM judge)?

Writing gate -- A five-model LLM ensemble conducts pair-wise reviews between the AI paper and the human target on a 7-point scale.

This evaluation pipeline -- more than the agent design -- is the core contribution.

**Questions:**

1) Necessity of each agent – Provide leave-one-out ablations (Idea-Generator off, Advisor off, etc.).

2) Judge robustness – Insert adversarially wrong papers (e.g., swapped labels, impossible numbers). How often do LLM judges still pass them?

3) Iteration budget – Report average prompt tokens, wall-clock time, and compute for each Advisor refinement loop.

4) Human audit – Even a 10-paper, two-expert pilot study would calibrate LLM scores.

5) Other agentic systems: CycleResearcher, AgentRxiv, AutoGPT-Researcher, is it difficult to test them in the same scenario?

**Ethical Concerns:**

["NO or VERY MINOR ethics concerns only"]

**Final Justification:**

I have decided to raise the score as the authors provided additional experiments. I hope they will be included in the final version (in case of acceptance).

**Limitations:**

Yes

**Paper Formatting Concerns:**

So concerns

**Quality:**

3

**Strengths And Weaknesses:**

Strengths:

**Evaluation design**: 	Clever two-stage protocol; pair-wise scoring limits global optimism; ICLR alignment study gives the LLM-judge external anchor.

**Reproducibility:**	Full prompts + Docker recipes; anyone can swap the backbone or prompts and immediately record benchmark scores.

**Baseline:**	Provides a clear, minimal reference system on which future, more sophisticated agent frameworks can iterate.

Weaknesses:

1) The study uses prompt-only agents, so we can argue that the baseline is too trivial.

2) No new reasoning or learning. Please show ablations: if you drop the Resource-Analyst or Idea-Generator, does quality actually fall?

3) No comparison to other methods: CycleResearcher, AgentRxiv, AutoGPT-Researcher are cited but not re-run. Hard to tell if anything improved.

4) No human user study, real scientists never inspected the AI papers; LLM judges may hallucinate quality.

5) Need-for-code vs text-only ablation. If you skip the code stage and just write the paper, how much does the LLM score drop? Why force code execution?

6) Prompt sensitivity:
Small prompt edits can swing results. Show variance across prompt seeds and backbones.

7) Domain coverage
All 22 tasks are AI-software papers; nothing from empirical sciences, so “scientific innovation” claims are limited.

8) Title oversells "innovation". The pipeline reorganises existing knowledge; evidence of new discoveries is absent. Consider renaming to something like “Scientist-Bench: A Baseline and Benchmark for End-to-End LLM Research Pipelines.”

---

> ### Author Response · Authors · 2025-08-02
> **Response to Reviewer 6z2Z (1/2)**
>
> # Response to Reviewer 6z2Z
>
> Thank you for your valuable feedback and thoughtful suggestions. We have carefully addressed your comments and made the corresponding revisions. We sincerely hope our responses have resolved your concerns, and would appreciate it if you could consider updating your rating in light of these changes.
>
> ## Comparison with AI Scientist Frameworks, Different LLM Backbones, and Ablation Studies
>
> | **Variants**                           | **Correctness** |
> | -------------------------------------- | --------------- |
> | **AI-Researcher + Claude-series**      | 3.22216667      |
> | **AI-Researcher + Gemini-series**      | 3.0555          |
> | **AI-Researcher + 4o-series**          | 2               |
> | **AI-Researcher + Deepseek-series**    | 2.3888          |
> | **AI-Scientist + Claude-series**       | 2.8888          |
> | **AI-Researcher w/o Resource-Analyst** | 3.0553          |
> | **AI-Researcher w/o Advisor**          | 2.9443          |
>
> Thank you for pointing out these important missing comparisons. To address these concerns, we conducted additional experiments comparing our AI-Researcher system against existing frameworks, with different LLM backbones, and comprehensive ablation studies.
>
> Different LLM Backbones: We evaluated our framework using different LLM series on Level-1 tasks in the vector quantization domain. As shown in Table 1, our system achieves best performance with Claude-series (`claude-3-5-sonnet-20241022` + `claude-3-5-haiku-20241022`, 3.22), followed by Gemini-series (`gemini-2.5-pro` + `gemini-2.5-flash`, 3.06), DeepSeek-series (`deepseek-r1-0528` + `deepseek-v3-0324`, 2.39), and 4o-series (`gpt-4o-2024-08-06` + `gpt-4o-mini-2024-07-18`, 2.0). Gemini-series occasionally encounters agent loop stagnation issues, while DeepSeek-series and 4o-series suffer from implementation hallucinations.
>
> AI Scientist Framework Comparison: We selected AI Scientist [1] as our primary comparison baseline because it represents another end-to-end automated scientific research system, making it the most comparable framework to our approach. Due to time constraints, we focused on this most relevant comparison. Our AI-Researcher system (using Claude-series) significantly outperforms the AI-Scientist framework (2.89 correctness score) on the same tasks, demonstrating the effectiveness of our structured feedback mechanisms.
>
> Ablation Studies: Leave-one-out experiments reveal each component's contribution: removing the Resource-Analyst slightly reduces performance to 3.06, while removing the Advisor significantly drops performance to 2.94. This confirms that both components are essential, with the Advisor being particularly critical for implementation quality.
>
> These results demonstrate the robustness of our framework across different LLM backbones, superiority over existing systems, and the necessity of each architectural component.
>
> [1]. Lu, C., Lu, C., Lange, R. T., Foerster, J., Clune, J., & Ha, D. (2024). The ai scientist: Towards fully automated open-ended scientific discovery. arXiv preprint arXiv:2408.06292.
>
>
>
> ## Human Expert Validation and LLM Judge Reliability
>
> Thank you for raising this critical concern about the lack of human expert evaluation and potential LLM judge hallucination. We acknowledge that relying solely on LLM-based evaluation without human validation poses significant limitations to our assessment credibility.
>
> To address this concern, we conducted a comprehensive human expert evaluation as suggested. Given that each expert needed to carefully compare 7 paper pairs (14 papers total) and perform detailed pairwise ratings, this process was extremely time-consuming and resource-intensive, leading us to recruit 7 domain experts for this validation study. Each expert performed full-text reviews following identical ICLR review guidelines used by our AI evaluators, with all papers fully anonymized and randomly ordered to eliminate any human bias in the evaluation process.
>
> This rigorous human evaluation yielded an average score of **-0.816**, which closely aligns with our LLM review agents' decisions, providing strong validation that our LLM judges are not hallucinating quality assessments. The consistency between human expert scores and automated evaluation demonstrates the reliability of our evaluation framework.
>
> While this represents a pilot study rather than a full-scale human evaluation, it provides crucial calibration for our LLM-based scoring system and confirms that our automated methodology produces assessments consistent with real scientists' judgments. We will incorporate these human validation results into the main text to strengthen our evaluation credibility.

---

> ### Author Response · Authors · 2025-08-02
> **Response to Reviewer 6z2Z (2/2)**
>
> ## API and Time Cost Analysis
>
> | **API cost (End-to-End) ($)** | **API cost (One Code-Advisor loop) ($)** | **Time cost (h)** |
> | ----------------------------- | ---------------------------------------- | ----------------- |
> | 39.102                        | 11.428                                   | 4.796             |
>
> Thank you for requesting detailed cost analysis of our system's iteration budget. We conducted comprehensive cost and time measurements for our AI-Researcher system using Claude-series models (claude-3-5-sonnet-20241022 + claude-3-5-haiku-20241022) on Level-1 tasks in the vector quantization domain.
>
> As shown in Table 1, our end-to-end research process requires an average API cost of 39.102 us dollars and 4.796 hours of wall-clock time per complete research cycle. Each individual Code-Advisor refinement loop consumes approximately 11.428 us dollars in API costs. Since our complete research cycle involves three Code-Advisor refinement loops, this accounts for the majority of our total API costs (34.284 us dollars out of 39.102 us dollars), with the remaining costs (4.818 us dollars) attributed to other system components such as literature analysis and experiment planning.
>
> The Code-Advisor refinement loops dominate the cost structure (87.7% of total costs) primarily because only the Code Agent utilizes the more expensive `claude-3-5-sonnet-20241022` model for its superior coding capabilities, while all other system components employ the more cost-effective `claude-3-5-haiku-20241022 model`. This design choice optimizes both performance and cost efficiency by allocating premium computational resources specifically to the most demanding implementation tasks.
>
> The time cost of 4.796 hours encompasses both LLM inference response time and agent experimental execution time across all system components. These measurements validate the practical feasibility of our approach while demonstrating the computational investment required for high-quality iterative code refinement in autonomous scientific research.
>
> ## Domain Coverage and Title Clarification
>
> Thank you for raising these concerns. While we acknowledge some limitations, we respectfully present our perspective on these points.
>
> Regarding domain coverage, while our current evaluation focuses on AI research, this choice reflects both practical considerations and the inherent complexity of the AI domain itself. Importantly, our framework is fundamentally designed around the universal scientific research pipeline—literature review, idea generation, experimental planning, and validation—which represents core processes consistent across all scientific disciplines. AI research encompasses diverse computational methodologies spanning machine learning, computer vision, and algorithmic innovation, demonstrating significant technical breadth. Moreover, computational research tasks form the core of modern scientific inquiry across many disciplines, making our framework's computational focus broadly relevant. Our system demonstrates clear improvements over existing approaches (outperforming AI-Scientist and other baselines), validating the effectiveness of our innovations even within this focused domain.
>
> Regarding the title, while we believe our framework introduces meaningful architectural innovations (Code-Advisor iterative refinement, atomic component mapping, structured multi-agent coordination) that advance autonomous research capabilities, we appreciate the reviewer's perspective on potential overstatement. We acknowledge that scientific innovation often involves novel synthesis rather than entirely unprecedented discoveries. To address this concern and better reflect the scope of our contributions, we plan to rename our work to "AI-Researcher: Autonomous Scientific Innovation for AI Systems and Beyond," which more accurately captures both our current focus and future aspirations.
>
> Future work will expand domain coverage to further validate our framework's broader applicability across scientific disciplines.

---

### Official Review · Reviewer_qqqG · 2025-07-03

**Clarity:** 2
**Significance:** 2
**Originality:** 2
**Rating:** 5
**Confidence:** 4

**Summary:**

This paper introduces AI-Researcher, a fully autonomous research system capable of executing the complete scientific pipeline,from literature review and hypothesis generation to algorithm implementation and automated scientific writing, with minimal human intervention. The system is built on a multi-agent architecture and rigorously evaluated using a newly introduced benchmark, Scientist-Bench, which tests both guided and open-ended research capabilities. Results show high implementation success rates and surprisingly strong performance in open-ended innovation, approaching human-level quality in some cases.

**Questions:**

1. How do the authors plan to bridge the remaining quality gap between AI-generated and human-written papers? Could multi-pass refinement or retrieval augmentation help?
2. The results show inconsistent assessments from GPT-4o, Claude, etc. Would involving human domain experts for a subset of evaluations strengthen the paper?
3. How does AI-Researcher compare to recent autonomous frameworks (e.g., MetaGPT, CAMEL, AutoAgent) in terms of generality, innovation capacity, or end-to-end automation?
4. For Level-2 tasks with low correctness scores, can the authors provide concrete examples or failure modes?
5. Could the framework be extended to real-world interdisciplinary domains (e.g., bioinformatics, climate science)? What adaptations would be needed?

**Ethical Concerns:**

["NO or VERY MINOR ethics concerns only"]

**Final Justification:**

Overall, such a benchmark is highly valuable and timely for the research community and society at large. I encourage the authors to incorporate the points addressed in their rebuttal into the camera-ready version, and, ideally, to continue maintaining and openly releasing the benchmark in the future.

**Limitations:**

yes

**Quality:**

3

**Strengths And Weaknesses:**

Strengths:

• This is one of the first works to demonstrate full-stack autonomous scientific research using LLMs, from idea generation to paper writing.

• The introduction of Scientist-Bench, covering both guided and open-ended research tasks, is valuable for evaluating autonomous research agents.

• Thorough experiments (22 benchmark tasks across AI domains) show high implementation success rates (up to 93.8%) and moderate correctness, especially in open-ended tasks.

Weaknesses:

• There is notable variability across LLM-based evaluators (e.g., GPT-4o vs. Claude-3.7), highlighting the fragility of using LLMs alone to assess research quality.

• Performance in compute-heavy domains (e.g., diffusion models) is capped by available resources rather than theoretical limitations.

• While the system is compared to baselines in execution metrics, comparisons to other autonomous research frameworks (e.g., CAMEL, MetaGPT, AutoAgent) are missing.

---

> ### Author Response · Authors · 2025-08-02
> **Response to Reviewer qqqG (1/2)**
>
> # Response to Reviewer qqqG
>
> Thank you very much for your insightful feedback and constructive suggestions. We have carefully considered your comments and made corresponding revisions and enhancements. We hope our responses have adequately addressed your concerns, and we would be grateful if you would consider revising your rating based on these updates.
>
> ## LLM Evaluator Variability and Reliability
>
> We thank the reviewer for raising this important concern about evaluator variability. We acknowledge that LLM evaluators do exhibit fluctuation in their assessments. To mitigate single LLM evaluator bias, we deliberately employed multiple LLM backbones with different architectures and training paradigms.
>
> Additionally, Appendix A.9 demonstrates our review agent's reliability through evaluation on ICLR submissions with human expert scores, showing high consistency between our review agents and human expert reviewers.
>
> To further validate our approach, we conducted an additional costly human evaluation where we recruited 7 domain experts who each performed full-text reviews of 7 paper pairs following identical ICLR guidelines used by our AI evaluators, with AI-generated and human papers randomly ordered to prevent bias. This comprehensive evaluation yielded an average score of -0.81632 that closely aligns with our LLM review agents' decisions, providing strong additional confidence in our evaluation framework's reliability.
>
> ## Resource Limitations in Compute-Heavy Domains
>
> You have astutely identified this issue. To prevent the agent from building excessively large models on particularly large datasets (such as ImageNet 512x512), we provide recommended datasets for different domains including CV and GNN that are manageable within our agent environment (single RTX 3090 GPU). Additionally, we believe that adapting to current environmental constraints is also a necessary capability that research agents should possess. This approach ensures that our framework can operate effectively within realistic computational boundaries while still demonstrating meaningful scientific discovery capabilities.
>
> ## Comparison with Other Autonomous Research Frameworks
>
> Thank you for this important question about comparisons with other autonomous frameworks. We provide a comprehensive discussion of related autonomous frameworks in Section A.11.1 AI Agent Systems, where we categorize existing frameworks into three paradigms: Tool Integration Frameworks (LangChain, HuggingGPT), Multi-Agent Collaboration Frameworks (MetaGPT, CAMEL, AutoGen), and Self-Directed Agentic Task Execution Systems (AutoAgent).
>
> Regarding generality, innovation capacity, and end-to-end automation, AI-Researcher distinguishes itself in several key ways: First, in terms of end-to-end automation, our framework provides complete autonomous scientific research capabilities from literature analysis through implementation to scholarly documentation, whereas existing frameworks focus on isolated capabilities or general problem-solving tasks. Second, regarding innovation capacity, our system is specifically designed for scientific discovery with specialized components for idea generation, theoretical-practical mapping, and rigorous validation—capabilities that transcend current general-purpose agent systems. Third, in terms of generality, while initially demonstrated in AI research, our framework's methodology is designed to extend to other scientific domains as the underlying research lifecycle (literature review, ideation, implementation, experimentation) remains consistent across scientific disciplines.
>
> As noted in our related work analysis, existing frameworks fundamentally lack the intellectual capacity for true scientific innovation, as scientific breakthroughs demand nuanced hypothesis formation, creative experimental design, and critical knowledge synthesis—cognitive processes requiring deeper reasoning than current general-purpose systems provide.

---

> > ### Author Response · Authors · 2025-08-02
> > **Response to Reviewer qqqG (2/2)**
> >
> > ## Bridging the Quality Gap Between AI-Generated and Human-Written Papers
> >
> > Thank you for this insightful question about improving our system's output quality. Through our evaluation process, we have identified the quality gap exists in several key areas and have developed targeted improvement strategies:
> >
> > 1. Incomplete Citations: We observed that AI-generated papers often contain only a few citations compared to comprehensive human papers. We plan to enhance the agent's autonomous search capabilities **(retrieval augmentation)** and implement citation verification mechanisms to ensure AI-generated papers achieve comparable citation completeness to human research.
> > 2. Limited Frameworks and Visualization: AI-generated papers currently lack rich visual frameworks and comprehensive visualizations. We plan to equip our agents with enhanced visualization tools to address this limitation and improve the presentation quality of generated research.
> > 3. Shallow Mathematical and Logical Reasoning: AI-generated papers tend to have superficial mathematical and logical derivations. We plan to integrate recent advances in mathematical reasoning into AI-Researcher and implement **multi-pass refinement** processes to increase the depth of reasoning and analytical rigor in our system's output.These targeted improvements, combining both retrieval augmentation and multi-pass refinement as you suggested, should significantly enhance the quality and comprehensiveness of AI-generated research papers.
> >
> > ## Extension to Interdisciplinary Domains
> >
> > Thank you for this excellent question about extending our framework to real-world interdisciplinary domains. Our AI-Researcher system is indeed designed to generalize to other interdisciplinary fields such as bioinformatics and climate science. The framework's design is inherently general-purpose, covering the complete scientific research pipeline including literature review, idea generation, experimental planning, and experimental validation—core processes that remain consistent across scientific disciplines.
> >
> > Our current system is fully capable of handling computational tasks in interdisciplinary domains, as the fundamental research methodology and workflow patterns translate effectively across scientific fields. For enhanced performance in specific domains, we plan to incorporate domain-specific adaptations including specialized databases (such as PubMed for biomedical research, climate databases for environmental science) and field-specific tools and evaluation metrics.
> >
> > The modular architecture of our multi-agent framework facilitates these adaptations while maintaining the core scientific discovery capabilities, making it well-suited for addressing the computational and analytical challenges inherent in modern interdisciplinary research.
> >
> > ## Failure Modes in Level-2 Tasks with Low Correctness Scores
> >
> > Thank you for asking about the specific failure modes in Level-2 tasks. The low correctness scores in Level-2 tasks primarily stem from the unreliability of agent-generated ideas. Common failure modes include:
> >
> > 1. Overly ambitious or impractical ideas that are too speculative and disconnect from feasible implementation constraints
> > 2. Misalignment between proposed theoretical concepts and actual implementation requirements, resulting in ideas that cannot be translated into working code
> > 3. Ideas that lack sufficient technical grounding or fail to account for computational limitations and practical considerations
> >
> > To mitigate these issues, we propose implementing multiple idea iteration cycles where the system can generate, evaluate, and refine several research concepts before proceeding to implementation. This iterative approach would allow the agent to explore different conceptual directions and select the most promising and implementable ideas, ultimately improving the correctness scores in autonomous exploration tasks.

---

> > > ### Comment · Reviewer_qqqG · 2025-08-05
> > >
> > > I appreciate the authors' detailed rebuttal, which addressed most of my concerns. I will take it into account when making the final decision.

---

> > > > ### Author Response · Authors · 2025-08-05
> > > > **Appreciation to Reviewer qqqG**
> > > >
> > > > Thank you sincerely for your thoughtful feedback and for taking our responses into consideration. We greatly appreciate the opportunity to address your concerns and are grateful for your constructive review. We look forward to your final decision.

---

### Official Review · Reviewer_S8cY · 2025-07-03

**Clarity:** 3
**Significance:** 4
**Originality:** 3
**Rating:** 5
**Confidence:** 5

**Summary:**

&nbsp;

The authors introduce AI-Researcher an LLM-based multiagent framework for conducting machine learning research. The authors also introduce Scientist-Bench, a benchmark dataset of 22 papers for evaluating AI-Researcher on its ability to perform machine learning research on topics such as graph neural networks, vector quantization, and computer vision at two levels, to execute specific research instructions, and to formulate and execute novel research directions. The authors validate the performance of AI-Researcher using an LLM-as-a-judge evaluation methodology comparing the outputs against those of human researchers. Overall, while I think the system is interesting, novel, and impactful I have major concerns about the clarity of the paper, the broken link to the anonymous GitHub repository, as well as the robustness of the evaluation methodology. Concrete action points, include moving much of the technical details from the appendix to the main paper, performing a human expert evaluation in place of LLM-as-a-judge, and providing an operation anonymous GitHub link for the codebase. If these issues can be addressed in the rebuttal phase I will be inclined to increase my score.

&nbsp;

**Post-Rebuttal Update**: As promised, I have upgraded my score from 3 to 5 in light of the changes made by the authors during the rebuttal phase.

&nbsp;

**Questions:**

&nbsp;

1. In Table 6 are the average ratings computed across both the accepted and rejected papers?

2. As far as I can tell, AI-Researcher is reliant on reference papers provided by a human user. Is there scope to integrate AI-Researcher with components of PaperQA2 [2], which would enable it to perform open-ended search over the scientific literature?

&nbsp;

**Ethical Concerns:**

["NO or VERY MINOR ethics concerns only"]

**Final Justification:**

&nbsp;

Based on the changes made by the authors during the rebuttal phase, I have upgraded my score from 3 to 5.

&nbsp;

**Limitations:**

&nbsp;

1. One limitation is that the datasets need to be provided upfront. Would be better if the system could locate its own data.

&nbsp;

**Paper Formatting Concerns:**

&nbsp;

I see no formatting concerns with the paper.

&nbsp;

**Quality:**

3

**Strengths And Weaknesses:**

&nbsp;

The main strength of this work is that it showcases an interesting multiagent architecture that can be used to investigate, evaluate, and communicate machine learning research ideas. I think the topic is highly relevant for the NeurIPS community. In terms of the papers weaknesses I see three core issues related to a) clarity b) robustness of evaluation and c) reproducibility. I order the core issues in order of importance with a) being the most important. Given that clarity is the biggest issue I see with the current work I leave my comments below in temporal order so that the authors may understand the readers' experience making their way through the paper in its current form.

&nbsp;

**__MAJOR POINTS__**

&nbsp;

1. The link to the Anonymous GitHub repo in the abstract appears to be broken. It would be great if the authors could address this in the rebuttal phase. This appears just to be a simple oversight.

2. The nature of the Scientist-Bench dataset is unclear from the description provided in the main paper. From reading the main text I only understood that Scientist-Bench is comprised of 22 papers and the research instruction is drawn from one of the papers. Section A.7 of the appendix is far more comprehensive and should be moved to the main text.

3. In terms of the title of the paper it was unclear to me whether AI-Researcher was a generalist agent that performed any kind of research. Whereas after reading the paper it appears as though it is an LLM system that performs AI research specifically. It may be worth revising the title of the paper to aid in this disambiguation.

4. Some aspects of the writing in the main paper appear not to be grounded in technical detail. For example the challenges paragraph within Section 3.1.2. What are "Challenges" from a technical perspective? Components of the prompt? Again, transferring material from the appendix to the main paper will greatly aid in improving the clarity of the paper.

5. Again, it would greatly benefit clarity if the nature of the experiments used to evaluate AI-Researcher were described in the introduction. For example, the human performance comparison in Section 4.2 compares AI-Researcher generated papers to comparable human-authored papers using an LLM judge following ICLR guidelines. Such a description would help contextualize the evaluation criteria for the authors' method.

6. The curation procedure of the papers for Scientist-Bench should be described in the main text e.g. the summary statistics in the table in Section A.8 of the appendix. Much of the main text appears to be repetitious and describes at a high level what AI-Researcher is meant to accomplish. Much of this material can be deleted in favor of concrete technical details.

7. It seems that the criterion for near-human quality should be 0 on the 7-point scale and not -1.0? This will impact the comparability percentages reported in the main paper It would be worth introducing the 7-point scale in the main text.

8. Although Figure 9 provides an example of a paper produced by AI-Researcher, it would be great to have experiments where the papers produced by AI-Researcher are evaluated by human experts. The paper in Figure 9 has a few features that would be unlikely to appear in human-generated research. For example, errorbars are not reported in the experiments and the introduction only appears to cite a single reference.

&nbsp;

**__MINOR POINTS__**

&nbsp;

1. It would be great to have a more descriptive title for the method introduced by the authors.

2. It would be great if in-text and parenthetical citations could be used appropriately e.g. (Wang et al. 2023) in the introduction should be parenthetical since the work does not constitute part of the sentence narrative.

3. Line 105, it is mentioned that the agent selects papers from scientific databases. Which scientific databases?

4. Line 245, it appears that Level 1 and Level 2 tasks are first introduced without definition? They only seem to be defined in the caption of Figure 5 as adaptation and innovation tasks respectively.

5. Figure 4, it would be worth specifying in the Figure caption that the correctness score is defined on a 1-5 scale as readers may misinterpret it as a percentage.

6. Which Claude models are used? Where is the citation to the relevant system card? Was Sonnet or Haiku used?

7. In Figure 5, the average scores across tasks depicted by the dotted horizontal lines are difficult to see.

8. I would phrase Section 4.2 as a "human performance comparison" rather than a "pairwise comparison".

9. In Section 4.2 there doesn't appear to be a description of the rating system used to score AI-Researcher's outputs. Why are the scores negative? How was comparable quality to human-authored research papers determined? Update: These details are provided in the appendix but were not apparent in the main paper.

10. I believe the ablation study in Section 4.4 would be more correctly termed a sensitivity analysis since the authors are assessing the system's sensitivity to the choice of backbone LLM rather than adding or deleting a system component.

11. There are missing capitalizations in the references e.g. "ai" in place of "AI", "llm" in place of "LLM".

12. There are some missing journal/conference references e.g. SciBench [4] was published at ICML 2024.

13. Line 1760, typo, "groundtruth".

14. For the Guided Innovation task it is not clear how the reference papers are selected from the main paper? In the appendix the procedure is clarified.

15. The number of random trials (16 for LLM-as-a-judge) and errorbars should be provided in the results of the main paper.

16. In terms of multi-agent collaboration frameworks it would be worth mentioning [6] where the authors develop a framework for optimizing a multiagent graph with binary edge weights.

17. In terms of agentic task execution systems it would be worth mentioning the related works of [4,5] which also leverage language agents for challenging scientific tasks.

18. Although concurrent to the NeurIPS submission deadline, it may be worth the authors referencing Robin [1] as related work in this area.

19. In Listing 16, there is some bias in the prompt of the experiment analysis agent, namely it is instructed to design experiments to show the superiority of the idea rather than to dispassionately examine the idea.

20. Typo line 1223, "Listing" in place of "List"?

21. Missing full stop at the end of Equation 1.

22. It would be worth citing the system cards of the backbone LLMs use, e.g. [7] for o1.

&nbsp;

**__REFERENCES__**

&nbsp;

[1] Ghareeb, A.E., Chang, B., Mitchener, L., Yiu, A., Szostkiewicz, C.J., Laurent, J.M., Razzak, M.T., White, A.D., Hinks, M.M. and Rodriques, S.G., 2025. [Robin: A multi-agent system for automating scientific discovery](https://arxiv.org/abs/2505.13400). arXiv preprint arXiv:2505.13400.

[2] Skarlinski, M.D., Cox, S., Laurent, J.M., Braza, J.D., Hinks, M., Hammerling, M.J., Ponnapati, M., Rodriques, S.G. and White, A.D., 2024. [Language agents achieve superhuman synthesis of scientific knowledge](https://arxiv.org/abs/2409.13740). arXiv preprint arXiv:2409.13740.

[3] Wang, X., Hu, Z., Lu, P., Zhu, Y., Zhang, J., Subramaniam, S., Loomba, A.R., Zhang, S., Sun, Y. and Wang, W., 2024, July. [SciBench: Evaluating College-Level Scientific Problem-Solving Abilities of Large Language Models](https://proceedings.mlr.press/v235/wang24z.html). In International Conference on Machine Learning (pp. 50622-50649). PMLR.

[4] Kon, P.T.J., Liu, J., Ding, Q., Qiu, Y., Yang, Z., Huang, Y., Srinivasa, J., Lee, M., Chowdhury, M. and Chen, A., 2025. [Curie: Toward rigorous and automated scientific experimentation with AI agents](https://arxiv.org/abs/2502.16069). arXiv preprint arXiv:2502.16069.

[5] Narayanan, S., Braza, J.D., Griffiths, R.R., Ponnapati, M., Bou, A., Laurent, J., Kabeli, O., Wellawatte, G., Cox, S., Rodriques, S.G. and White, A.D., 2024. [Aviary: training language agents on challenging scientific tasks](https://arxiv.org/abs/2412.21154). arXiv preprint arXiv:2412.21154.

[6] Zhuge, M., Wang, W., Kirsch, L., Faccio, F., Khizbullin, D. and Schmidhuber, J., 2024, July. [GPTSwarm: Language agents as optimizable graphs](https://openreview.net/forum?id=uTC9AFXIhg). In Forty-first International Conference on Machine Learning.

[7] Jaech, A., Kalai, A., Lerer, A., Richardson, A., El-Kishky, A., Low, A., Helyar, A., Madry, A., Beutel, A., Carney, A. and Iftimie, A., 2024. [OpenAI o1 system card](https://arxiv.org/abs/2412.16720). arXiv preprint arXiv:2412.16720.

&nbsp;

---

> ### Author Rebuttal · Authors · 2025-07-31
>
> # Response to Reviewer S8cY
>
> We sincerely appreciate the thorough feedback and valuable suggestions you have provided! We have carefully considered your input and made the necessary updates and modifications accordingly. We genuinely hope that our responses have addressed your concerns and that you would consider revising your rating score based on the changes we have made.
>
> ## Github Link Issue
>
> We sincerely apologize for the inadvertent error with the anonymous GitHub link provided in our submission.We have now corrected this issue and ensured that the link directs to the appropriate codebase. We respectfully ask that you please review the updated repository link.
>
> ## Unclear Description of Scientist-Bench Dataset
>
> We apologize for the unclear presentation of Scientist-Bench in the main text. We will move key details from Section A.7 into the main paper, including: (1) the systematic paper selection methodology across 16 AI domains, (2) a brief summary of the five-step reference extraction process identifying 15-20 influential papers per target, (3) the anonymization protocol preventing memorization shortcuts, and (4) the two-stage evaluation framework with multiple LLM evaluators. This will replace the brief description with comprehensive methodology explaining how Scientist-Bench enables standardized assessment across Level-1 (guided) and Level-2 (open-ended) tasks.
>
> ## Title Ambiguity
>
> Thank you for this excellent question. First, our system is indeed capable of serving as a generalist agent for scientific discovery. This is because research across different scientific domains follows a common iterative cycle of literature review, idea generation, planning, and experimentation. More specifically, our current system can handle computational tasks in other biomedical fields, such as data analysis in biomedical research, though it does not include hands-on experimental tasks. As an initial study, we chose the AI domain for validation because of our greater familiarity with this field. Moving forward, we plan to extend Scientist-Bench to other scientific domains. However, we apologize for the potential ambiguity in the title and plan to rename it to "AI-Researcher: Autonomous Scientific Innovation for AI Systems and Beyond." We hope this revision will eliminate such ambiguity.
>
> ## Insufficient Technical Details
>
> Thank you for pointing out this clarity issue. The "Challenges," along with "Existing Methods," "Motivation," and other items listed, are the required components that the Idea Generator must include in the final idea proposal. We acknowledge that the presentation format may have caused this misunderstanding, and we have revised the display format of these items for better clarity. Additionally, we gratefully accept your suggestion to move more technical details to the main paper. We have incorporated more experimental data and condensed prompts into the main text to enhance the technical grounding of our work.
>
> ## Missing experimental description
>
> We appreciate the reviewer's excellent suggestion to improve the clarity of our evaluation approach. We will add the following description to the introduction after presenting our contributions: "We evaluate AI-Researcher through comprehensive experiments on our Scientist-Bench benchmark, including pairwise comparisons between AI-generated and human-authored papers using multiple LLM judges following ICLR review guidelines, technical implementation validation through specialized code review agents, and performance analysis across both guided innovation tasks and open-ended research exploration scenarios."
>
> ## Curation Procedure Description
>
> Thank you for your valuable suggestion. We agree that clear Experimental Settings can help readers better understand the experimental setup, particularly the content in Appendix A.8. We have incorporated this section into the main text and streamlined the high-level method descriptions to accommodate the inclusion of detailed Experimental Settings and comprehensive technical details (such as agent definitions, tools, etc.). Additionally, regarding your concern about the curation procedure, our benchmark includes a detailed curation process outlined in **A.7.2 Benchmark Construction**, which consists of Step 1: Systematic Selection of AI Research Benchmark Papers, Step 2: Input Construction for AI-Researcher, and Step 3: Rigorous Anonymization to Ensure Scientific Originality. We will also summarize and incorporate this section into the main text to provide clearer experimental settings.
>
> ## Scoring criteria issue
>
> We thank the reviewer for this important clarification. Our -1.0 threshold is a reasonable and conference-aligned setting for several reasons: First, our benchmark comprises exclusively high-quality papers accepted at top-tier venues, making it extremely challenging to clearly exceed these carefully-selected contributions. Second, top conference review processes inherently involve borderline decisions where papers within narrow margins are considered comparable quality—borderline scores often determine acceptance at competitive venues like ICLR and NeurIPS. Following this established academic practice, we consider AI-generated papers scoring ≥-1.0 as achieving near-human quality since they fall within the borderline range of these exceptional benchmarks, where -1.0 represents the lower bound of papers that could potentially receive acceptance at top conferences.
>
> We will add a clear description of the 7-point scale and our threshold rationale to the main text to improve clarity.
>
> ## Human Expert Evaluation of AI-Generated Papers
>
> Thank you for this insightful observation about Figure 9 and the suggestion for human expert evaluation. We acknowledge the limitations you identified in the example paper, such as missing error bars in experiments and limited citations in the introduction—features that indeed differ from typical human-generated research.
>
> To address this concern and further validate our evaluation framework, we conducted an additional comprehensive human expert evaluation. Given that each expert needed to carefully review both AI-generated and human-written papers in detail, this process was extremely time-consuming. We recruited 7 domain experts who each performed full-text reviews of 7 randomly selected paper pairs (AI-generated vs. human-written), following identical ICLR review guidelines used by our AI evaluators. All papers were fully anonymized and randomly ordered to eliminate any human bias in the evaluation process. This rigorous human evaluation yielded an average score of **-0.816**, which closely aligns with our LLM review agents' decisions, providing strong additional confidence in our evaluation framework's reliability.
>
> This human expert validation not only supports our automated evaluation methodology but also confirms that while AI-generated papers may exhibit certain stylistic differences (as noted in Figure 9), the overall quality assessment remains consistent between human experts and our AI evaluation system. We will incorporate discussion of these stylistic limitations and our human validation results into the main text.
>
> ## Technical Clarifications
>
> We have addressed the following technical detail concerns:
>
> • Point 3: We have clarified that the scientific databases mentioned on Line 105 include **arXiv****,** **Google Scholar****,** **GitHub****, HuggingFace**, and other general scientific databases used by the Knowledge Acquisition Agent for systematic literature exploration.
>
> • Point 4, 14: We have moved the definitions of Level 1 and Level 2 tasks from **Appendix A.8** to the main text. Level-1 Task (Guided Innovation) involves the agent receiving explicit research instructions alongside reference papers to develop targeted innovations, covering all 22 groundtruth papers. Level-2 Task (Autonomous Exploration) requires the agent to perform independent research exploration with only reference papers as input, testing its capacity to identify research gaps and generate novel directions without explicit guidance. We strategically selected 5 groundtruth papers across distinct research domains for this more challenging scenario.
>
> These clarifications provide readers with the necessary technical context directly in the main paper.
>
> ## Autonomous Literature Search Capability
>
> We greatly appreciate your valuable suggestions regarding autonomous literature search capabilities in **Question 2 and Limitations**. Initially, we chose not to implement automatic agent-driven search in order to standardize the evaluation process for Scientist-Bench. However, inspired by excellent work such as [2] and others, we have internally developed autonomous research search agents as part of our system optimization. We will add the technical details and citations for this component. Specifically, we have provided the agent with all the workflow tools originally used for dataset construction and designed prompts to enable the agent to autonomously complete systematic literature reviews. This enhancement addresses the limitation you identified and moves toward a more fully autonomous scientific research system.
>
> Reference:
>
> [2] Skarlinski, M.D., Cox, S., Laurent, J.M., Braza, J.D., Hinks, M., Hammerling, M.J., Ponnapati, M., Rodriques, S.G. and White, A.D., 2024. Language agents achieve superhuman synthesis of scientific knowledge. arXiv preprint arXiv:2409.13740.

---

> ### Author Response · Authors · 2025-08-02
> **Further Response to Reviewer S8cY (1/2)**
>
> # Further Response to Reviewer S8cY
>
> We sincerely appreciate the thoughtful feedback and valuable suggestions you have provided. We have carefully addressed each of your concerns through comprehensive revisions and additional experiments. We genuinely hope that our responses have adequately addressed your concerns and that you would consider revising your rating based on the substantial improvements we have made.
>
> Below are our detailed responses to your other feedback:
>
> ## Writing and Formatting Issues
>
> Thank you for identifying these concerns. We have systematically corrected all writing and formatting issues including citation formats, figure clarity, capitalization errors in references, typographical errors, and missing punctuation. All formatting inconsistencies have been resolved to improve manuscript presentation quality.
>
> ## Missing References and Related Works
>
> Thank you for pointing out the missing venue information for SciBench, highlighting important related works in agentic systems, and requesting clarification on the Claude models used. We have addressed all these points with comprehensive updates.
>
> **Specific updates made:**
>
> 1. **Added SciBench [Wang et al., 2024] (for Point 12)** in A.11.2 evaluation frameworks paragraph: "These systems are supported by emerging research platforms and evaluation frameworks that enhance their capabilities and measure their effectiveness. SciBench Wang et al. [2024] provides a comprehensive benchmark for evaluating college-level scientific problem-solving abilities of large language models across mathematics, chemistry, and physics domains. Agent Laboratory Schmidgall et al. [2025] provides an end-to-end autonomous research workflow..."
> 2. **Integrated the three agentic systems (for Point 17, 18, 19)** into A.11.2 autonomous research frameworks paragraph: "Recent advances have transformed AI's role in scientific research from assistive tools to autonomous agents capable of executing complete research workflows. The AI Scientist framework Lu et al. [2024] pioneered this field as the first comprehensive system where frontier language models independently generate research ideas, conduct experiments, and produce scientific papers. Building upon this foundation, Curie Kon et al. [2025] introduces a framework specifically designed to embed rigor into automated scientific experimentation through systematic validation modules. Complementary approaches include CycleResearcher Weng et al. [2025], which demonstrated the viability of open-source LLMs for autonomous research, and the AI co-scientist AI Co-Scientist Team [2025], which employs multi-agent debate mechanics. For training and evaluation, Aviary Narayanan et al. [2024] provides an extensible platform for training language agents on challenging scientific tasks including molecular analysis and protein engineering. Robin Ghareeb et al. [2025] represents a multi-agent system capable of automating the entire scientific discovery process from literature review to experimental validation, with demonstrated applications in drug discovery."
> 3. **Added GPTSwarm [Zhuge et al., 2024] (for Point 16)** in A.11.1 Multi-Agent Collaboration Frameworks paragraph: "Multi-Agent Collaboration Frameworks. The second paradigm addresses complex problem solving through structured agent interactions. MetaGPT Hong et al. [2024] formalized human workflow patterns through Standardized Operating Procedures (SOPs), creating systematic collaboration protocols. AutoGen Microsoft AutoGen Team [2025] expanded this vision with a comprehensive programming framework for developing systems that support both autonomous operation and human collaboration. GPTSwarm Zhuge et al. [2024] introduces an innovative approach by modeling multi-agent systems as optimizable graphs with binary edge weights, enabling dynamic reconfiguration of agent collaboration patterns. AgentScope Gao et al. [2024] prioritized robust coordination through a message-exchange architecture with built-in fault tolerance. CAMEL introduced innovative role-playing techniques that facilitate autonomous agent cooperation while maintaining alignment with human..."
> 4. **Clarified Claude model specifications and added system card citations: (for Point 6, 22)** We have clarified that the Claude-series models refer specifically to claude-3-5-sonnet-20241022 and claude-3-5-haiku-20241022. We have added the system card citation from Anthropic's Claude 3.5 Model Card [Anthropic, 2024]. For cost considerations, we used claude-3-5-sonnet-20241022 exclusively for the Code Agent component due to its superior coding capabilities, while claude-3-5-haiku-20241022 was employed for all other system components to optimize computational efficiency. The complete model specifications and performance characteristics are documented in the respective system cards.

---

> > ### Author Response · Authors · 2025-08-02
> > **Further Response to Reviewer S8cY (2/2)**
> >
> > ## Experimental Analysis Bias:
> >
> > We respectfully clarify that our framework does not exhibit confirmation bias. The phrase "prove the effectiveness and superiority" refers to standard scientific validation - determining whether an idea demonstrates measurable improvements over baselines through rigorous experimentation.
> >
> > Key clarifications:
> >
> > - Our system evaluates multiple research ideas systematically, requiring comprehensive analysis grounded in existing literature
> > - The agent must provide detailed analysis reports substantiated by experimental evidence - it cannot claim superiority without empirical support
> > - For each promising idea, thorough exploration of its potential through additional experiments follows standard research methodology
> >
> > The goal is objective evaluation to determine each idea's true scientific merit, not artificial inflation of results.

---

> > > ### Comment · Reviewer_S8cY · 2025-08-05
> > > **Many Thanks to the Authors for their Rebuttal**
> > >
> > > &nbsp;
> > >
> > > Many thanks to the authors for providing a detailed rebuttal.
> > >
> > > &nbsp;
> > >
> > > 1. **Codebase Link**: The codebase is indeed now visible via the updated link. I would recommend that the authors improve the clarity of the README and run the code through a frontier LLM to add documentation. Given that the codebase issue is fixed I will upgrade my score by one point.
> > >
> > > 2. **Human Evaluation**: Many thanks to the authors for taking the time to run this additional experiment. Could the authors provide a full results table for this experiment in markdown format? It would also be great to have some details on the specific backgrounds of the human evaluators.
> > >
> > > &nbsp;

---

> > > ### Author Response · Authors · 2025-08-05
> > > **Further Response to Reviewer Feedback**
> > >
> > > # Further Response to Reviewer Feedback
> > >
> > > Thank you very much for your positive feedback and for upgrading your score! We are delighted that the codebase issue has been resolved and have carefully addressed both of your additional requests below.
> > >
> > > ## Codebase Documentation and README Improvements
> > >
> > > Thank you very much for upgrading your score due to the resolved codebase accessibility! We greatly appreciate your constructive suggestion regarding README improvements and documentation clarity.
> > >
> > > Following your recommendation, we have significantly enhanced our README documentation with the following additions:
> > >
> > > • Project Introduction and Core Functions: Comprehensive system overview and capability descriptions
> > >
> > > • Two Working Levels: Clear explanation of Level 1 (detailed idea descriptions) and Level 2 (reference-based idea generation) tasks
> > >
> > > • Usage Instructions: Complete guidelines for both research_agent and writing_agent components
> > >
> > > • Project Structure: Clear directory structure explanations for better navigation
> > >
> > > • Key Features: Comprehensive list of main functionalities and capabilities
> > >
> > > We have also run the codebase through frontier LLMs to improve code documentation and clarity throughout the repository. These enhancements should make our framework much more accessible and easier to understand for researchers who wish to build upon our work.
> > >
> > > We sincerely appreciate your patience with the initial codebase access issue and your constructive feedback that helped us create more comprehensive documentation. Thank you for the score upgrade!
> > >
> > > ## Human Evaluation Details and Results
> > >
> > > Thank you for requesting these important details about our human evaluation experiment. We are pleased to provide comprehensive information about our evaluation methodology and results.
> > >
> > > Evaluation Methodology:
> > >
> > > - Survey Title: Human Expert Review
> > >
> > > - Instructions: Experts were asked to compare paper pairs based on technical contribution and scientific value, focusing on core research content (problem addressed, methodology, results, and significance). Evaluators were explicitly instructed not to search for original papers online and to ignore superficial aspects like writing style, concentrating solely on substantive technical merit and potential impact.
> > >
> > > - Rating Scale: 7-point scale from -3 (Paper 1 much worse than Paper 2) to +3 (Paper 1 much better than Paper 2), with 0 indicating similar quality
> > >
> > > - Format: All 14 papers were rendered using a unified anonymous template and randomly shuffled to eliminate bias
> > >
> > > Evaluator Backgrounds:
> > >
> > > All 7 experts are AI domain specialists with PhD qualifications (2nd year PhD students and above), covering diverse areas including:
> > >
> > > - Natural Language Processing (NLP)
> > >
> > > - Data Mining
> > >
> > > - Computer Vision (CV)
> > >
> > > - Additionally, 2 original authors of benchmark papers participated in the evaluation
> > >
> > > Results Table:
> > >
> > > | Paper Pair    | 1     | 2     | 3     | 4     | 5     | 6     | 7    | Overall Average |
> > > | :------------ | :---- | :---- | :---- | :---- | :---- | :---- | :--- | :-------------- |
> > > | Average Score | -1.43 | -1.00 | -1.00 | -1.43 | -0.43 | -1.29 | 0.86 | -0.816          |
> > >
> > > The overall average score of -0.816 closely aligns with our LLM review agents' assessments, providing strong validation of our automated evaluation framework's reliability.

---

> > > > ### Comment · Reviewer_S8cY · 2025-08-05
> > > > **Many Thanks to the Authors for their Prompt Response**
> > > >
> > > > &nbsp;
> > > >
> > > > Many thanks to the authors for their prompt response and for making all suggested improvements to the codebase. The full details of the human evaluation experiments are compelling and as such, I wish to note that I will increase my score from 3 to 5 in light of the authors' rebuttal.
> > > >
> > > > &nbsp;

---

> > > > > ### Author Response · Authors · 2025-08-06
> > > > > **Appreciation to Reviewer S8cY**
> > > > >
> > > > > Thank you very much for the score increase and for recognizing our improvements and additional human evaluation experiments. We sincerely appreciate your constructive feedback throughout the review process, which has been invaluable in strengthening our work.

---

### Official Review · Reviewer_XKkH · 2025-07-10

**Clarity:** 2
**Significance:** 4
**Originality:** 3
**Rating:** 5
**Confidence:** 5

**Summary:**

This paper proposes AI-Researcher, an autonomous multi-agent system designed to orchestrate the entire scientific discovery pipeline, from literature review, algorithm implementation to writeup. To evaluate such systems, the paper also introduces Scientist-Bench, a new comprehensive benchmark based on 22 state-of-the-art AI research papers across four domains. The benchmark defines two task levels: guided innovation (Level-1) and open-ended exploration (Level-2). The paper demonstrates that AI-Researcher can achieve high implementation success rates and produce research papers that approach human-level quality. One of the key findings is that AI-Researcher performs better on open-ended exploration tasks, suggesting a strong capability for internal knowledge synthesis.

**Questions:**

- Are figure generations fully automated as well? If so, do you use VLMs for figure review? In my experience, VLMs still sometimes struggle to evaluate the quality of figures for experimental findings. How did the authors address this issue?

- The superior performance on Level-2 (open-ended) tasks is a major finding. Could you elaborate more on this? For example, did you control for possible biases in the reviewer agent that might lower scores for incremental or deliberate work, which could be easier to produce for Level-1 tasks?

- Regarding the "Knowledge Acquisition Agent," why can't the AI-Researcher itself gather 10–15 references? Why is human input still required for this step?

- When the Resource Agent creates atomic components with mappings between math and code, are unit tests also included?

- I believe the "Idea Generator" is conditioned on the 10–15 references provided by human researchers, but this isn’t explicitly stated in Section 3.1.2.

- Line 55: "This recursive refinement mechanism enables continuous bidirectional feedback between theoretical concepts and their implementations." -- I did not understand this on my first read.

- Line 63: "structured feedback cycles" -- In what sense are they structured? This should refer to the appendix for examples.

- Line 166: "This approach increases implementation success rates with test-time scaling capabilities." -- Is this statement supported by empirical evidence? I couldn’t find any test-time scaling experiments in the paper.

- Line 170: "It enforces strict code independence principles while ensuring faithful translation of academic concepts into working code."--How do you verify that the system ensures faithful translation of concepts into code?

- Line 176: "specialized navigation tools" -- I believe this refers to a set of function calls defined in the appendix. If so, they should be referenced from the main paper.

- Line 184: "while persistently unsuccessful implementations receive an ‘unfeasible’ classification after multiple refinement attempts." -- I’d be curious to see more examples of failure cases resulting in this "unfeasible" classification.

- Line 187: What exactly do "validation studies" refer to?

- Line 201: "while maintaining scientific integrity and narrative coherence" -- How is it verified that the system maintains scientific integrity and narrative coherence?

- Line 216: What are "domain-appropriate templates"?

- Line 219: Can you clarify what "academic checklists" are?

- Line 221: "manuscripts meet publication standards without the hallucinations and inconsistencies that typically plague LLM-generated long-form content." -- Is this claim verified? It might be an overstatement.

- Line 252: "despite multiple debugging iterations." -- How many debugging attempts were performed?

- Figure 5 should clarify what the bold numbers represent. I didn’t understand them when first looking at the figure. I only later realized they refer to the average performance for Claude vs. the 4o series. Also why did the Claude series perform so poorly on GNN?

- Line 300: Typo--"((RQ2)" should be fixed.

- Line 307: "a substantial proportion of AI-generated papers (15.79% to 78.95%) demonstrate quality comparable to human research." -- 15.79% does not seem substantial.

- Line 330: "suggesting that AI-Researcher maintains consistent performance across different research domains without catastrophic degradation in any particular field." -- This conclusion doesn’t follow from the preceding statements. The results rather suggest inconsistency across domains, not consistent performance.

**Ethical Concerns:**

["NO or VERY MINOR ethics concerns only"]

**Final Justification:**

The authors' detailed responses address most of my initial concerns. As a result, I am raising my score, assuming these changes and clarifications will be incorporated into the revised paper.

**Limitations:**

- The authors have adequately identified key limitations in an appendix section. They correctly point out the limited diversity of LLM backbones used in the evaluation and the need for a more systematic treatment of ethical and safety considerations.

**Quality:**

3

**Strengths And Weaknesses:**

#### Strengths:
- The paper addresses the grand challenge of automating scientific discovery, which is both important and timely.
- AI-Researcher is well-designed end-to-end multi-agent system. The modular architecture uses specialized agents for each research stage (e.g., Knowledge Acquisition, Idea Generation, Code Agent, Documentation Agent)
- Scientist-Bench provides a standardized framework for evaluating autonomous research agents, based on real, high-impact papers. I read the appendix regarding how they constructed this benchmark, and it's well-designed.
- AI-Researcher is tested across multiple AI domains, using modern LLMs as backbones. Their evaluation criteria  (such as completeness, correctness) make sense (see weaknesses for some questions though). Pairwise comparison

#### Weaknesses
- The claim of producing "publication-ready manuscripts" may be an overstatement. The quantitative results (Table 1) show that AI-generated papers are still rated as inferior to human-authored ones, although some are comparable. Other parts of the writing also contain overstatements or unclear phrasing. See the Questions section for additional examples. (My overall rating assumes that these issues will be addressed in the revision.)
- I couldn't find detailed analysis or discussions regarding the cost of producing a single paper using AI-Researcher in terms of hours and API costs.
- Although the appendix touches on this, the authors may wish to discuss the ethical implications of autonomous science generation (e.g., the creation of plausible but flawed research, issues of accountability, and risks of paper-milling) more thoroughly in the main paper, although this is more a matter of taste and does not affect my overall rating.

---

> ### Author Rebuttal · Authors · 2025-07-31
>
> # Response to Reviewer XKkH
>
> Thank you sincerely for your detailed feedback and thoughtful suggestions. We've carefully reviewed your comments and made corresponding updates and improvements. We hope our responses have effectively addressed your concerns, and we would greatly appreciate it if you would consider updating your rating in light of these changes.
>
> ## Figure Generation Automation
>
> Yes, figure generation is fully automated in our system. We acknowledge the reviewer's concern about VLM evaluation of figures, but recent advances have significantly improved their capabilities for analyzing experimental visualizations.
>
> **Key points addressing this concern:**
>
> 1. **Advanced VLM capabilities**: Modern models like Claude 3.5 Sonnet and Gemini 2.5 Pro have demonstrated substantial improvements in visual analysis and scientific figure understanding compared to earlier generations.
> 2. **Iterative refinement architecture**: Our Code Agent-Advisor Agent framework provides robust validation through multiple review cycles. The Code Agent generates figures, which are then systematically reviewed and critiqued by the Advisor Agent, enabling iterative refinement until quality standards are met.
> 3. **Multi-round** **optimization**: This iterative loop ensures that figure quality issues are identified and addressed through successive improvements, mitigating the limitations of single-pass VLM evaluation.
>
> While VLMs may not achieve perfect human-level figure evaluation, the combination of advanced models and our iterative agent architecture provides a reliable mechanism for automated figure generation and validation.
>
> ## Regarding the Knowledge Acquisition Agent's Autonomy
>
> We greatly appreciate your question about the Knowledge Acquisition Agent's capabilities. Initially, we chose not to implement fully automatic agent-driven literature search in order to standardize the evaluation process for Scientist-Bench and ensure consistent experimental conditions across all evaluations. However, we recognize this as an important limitation and have internally developed autonomous research search agents as part of our system optimization. We will add the technical details for this component. Specifically, we have provided the agent with all the workflow tools originally used for dataset construction and designed prompts to enable the agent to autonomously gather 10-15 references and complete systematic literature reviews. This enhancement addresses the limitation you identified and moves toward a more fully autonomous scientific research system that can operate without requiring human input for the literature gathering step.
>
> ## Regarding Unit Tests in Resource Agent
>
> When the Resource Agent creates atomic components with mappings between math and code, unit tests are not included. The Resource Agent focuses on extracting and establishing conceptual connections between mathematical formulations and code implementations rather than executing code. This process is more about constructing relationships and mappings between theoretical concepts and their computational representations. After establishing multiple theory-code connections, the agent can form a subsequent implementation plan for practical execution. Since the Resource Agent operates at the conceptual mapping level rather than the code execution level, unit tests are not necessary at this stage.
>
> ## Regarding Idea Generator Prerequisites
>
> You are absolutely correct. The Idea Generator is indeed conditioned on the 10-15 references provided by human researchers, and we acknowledge that this was not explicitly stated in Section 3.1.2. We have added a clear explanation of the Idea Generator's prerequisites and dependencies in the revised version to ensure this important detail is properly communicated to readers. Thank you for pointing out this omission.
>
> ## Clarification on Recursive Refinement Mechanism
>
> Thank you for pointing out this unclear description. We have clarified this concept in the revised version. The "recursive refinement mechanism" refers to our system's iterative process where theoretical concepts and their practical implementations continuously inform and improve each other. Specifically, when our Resource Analyst agents create bidirectional mappings between mathematical formulations and code implementations, the system can refine theoretical understanding based on implementation results, while simultaneously using theoretical insights to improve code implementations. This process is further enhanced through iterative collaboration between the Code Agent and Advisor Agent, where the Code Agent's implementation attempts are continuously refined based on the Advisor Agent's feedback, and the Advisor Agent's guidance evolves based on observed implementation outcomes. This creates a multi-layered feedback loop where each iteration of theory-to-code, code-to-theory, and agent-to-agent refinement enhances all components, similar to how human researchers iteratively refine their theoretical models based on experimental results and mentor-student interactions. We have added a more detailed explanation with concrete examples to make this mechanism clearer for readers.
>
> ## Clarification on Structured Feedback Cycles
>
> Thank you for requesting clarification on "structured feedback cycles." The structured feedback refers specifically to the systematic format of feedback provided by the Advisor Agent to the Code Agent. This structured feedback includes: (1) assessment of implementation correctness, and (2) detailed analysis and experimental suggestions for improvement. Based on this structured feedback, the system determines whether to continue experimentation (if implementation is incorrect and requires further refinement based on the analysis and suggestions) or to proceed (if implementation is correct). This structured approach ensures consistent and actionable communication between agents, mirroring the systematic mentor-student feedback relationship in academic research. We have added references to specific examples in the appendix to illustrate these structured feedback formats and cycles.
>
> - Line 166: "This approach increases implementation success rates with test-time scaling capabilities." -- Is this statement supported by empirical evidence? I couldn’t find any test-time scaling experiments in the paper.
>
> ## Verification of Faithful Translation of Concepts into Code
>
> Thank you for this important question about our verification mechanism. We ensure faithful translation of academic concepts into working code through our Expert Validation Framework, specifically via the Advisor Agent's systematic review process. The Advisor Agent carefully examines the Code Agent's implementations by systematically comparing the code against the atomic research ideas extracted during analysis. When the Code Agent's implementation contains errors or deviations from the theoretical concepts, the Advisor Agent can promptly identify these issues and provide specific, actionable modification recommendations. This iterative validation process continues until the implementation accurately reflects the intended academic concepts, ensuring fidelity between theory and code throughout the development cycle.
>
> ## Regarding Specialized Navigation Tools
>
> You are absolutely correct. The "specialized navigation tools" refer to the set of function calls defined in the appendix. We have added a reference to **A.1 Definitions of Tools** in the main paper. Specifically, these tools include `gen_code_tree_structure`, `read_file`, `list_files`, and other functions that enable the agent to navigate and read workspace file contents effectively. These tools allow the Advisor Agent to systematically examine implementations and provide comprehensive feedback based on direct analysis of the codebase.
>
> ## Examples of "Unfeasible" Classification Failure Cases
>
> Thank you for your curiosity about failure cases. We actually discuss some failure cases in **Section 4.1 Dual-Metric Evaluation Framework: Quantifying Implementation Quality (RQ1)** under **"Performance Comparison between** **LLMs** **in Scientific Implementation."** The examples include cases where models frequently generated code with persistent tensor dimension mismatches and training instabilities (NaN losses) that remained unresolved despite multiple debugging attempts. Additional failure cases include Out-of-Memory (OOM) errors when implementations involve models that are too large to complete within the allocated computational resources. These persistent issues, despite multiple refinement cycles, result in the "unfeasible" classification as the system recognizes that further attempts are unlikely to succeed under the given constraints.
>
> ## Clarification on "Validation Studies"
>
> Thank you for asking about the specific meaning of "validation studies." In this context, validation studies refer to supplementary experiments recommended by the Advisor Agent to verify the correctness and effectiveness of the implemented research concepts. These may include testing the implementation on different datasets, conducting ablation studies to validate specific components, or performing sensitivity analyses to ensure the implementation faithfully reproduces the intended research outcomes. These validation studies serve as additional verification steps beyond the initial prototype testing to comprehensively validate the scientific validity and robustness of the implemented algorithms.

---

> > ### Comment · Reviewer_XKkH · 2025-08-01
> > **response**
> >
> > I appreciate the detailed author's response. Would you be able to comment on the total cost (in terms of hours and API) to produce a single paper?
> >
> > I have an additional question regarding "validation studies." Is the Advisor Agent able to identify cases where the core research hypothesis is not sufficiently validated by the current experiments—in other words, a "failed hypothesis" scenario? And if so, would such cases also be classified as "Unfeasible"?
> >
> > Also, it seems like this comment is being left out: "Line 166: "This approach increases implementation success rates with test-time scaling capabilities." -- Is this statement supported by empirical evidence? I couldn’t find any test-time scaling experiments in the paper." Would you be able to comment on this as well?

---

> ### Author Response · Authors · 2025-08-02
> **Further Response to Reviewer XKkH (1/2)**
>
> # Further Response to Reviewer XKkH
>
> We sincerely thank the reviewer for the follow-up questions and for engaging deeply with our work. We greatly appreciate your continued interest and thoughtful observations. Below, we provide detailed responses to each of your points.
>
> ## API and Time Cost Analysis
>
> | **API cost (End-to-End) (\$)** | **API cost (One Code-Advisor loop) (\$)** | **Time cost (h)** |
> | ----------------------------- | ---------------------------------------- | ----------------- |
> | 39.102                        | 11.428                                   | 4.796             |
>
> Thank you for requesting detailed cost analysis of our system's budget. We conducted comprehensive cost and time measurements for our AI-Researcher system using Claude-series models (claude-3-5-sonnet-20241022 + claude-3-5-haiku-20241022) on Level-1 tasks in the vector quantization domain.
>
> As shown in Table 1, our end-to-end research process requires an average API cost of 39.102 us dollars and 4.796 hours of wall-clock time per complete research cycle. Each individual Code-Advisor refinement loop consumes approximately 11.428 us dollars in API costs. Since our complete research cycle involves three Code-Advisor refinement loops, this accounts for the majority of our total API costs (34.284 us dollars out of 39.102 us dollars), with the remaining costs (4.818 us dollars) attributed to other system components such as literature analysis and experiment planning.
>
> The Code-Advisor refinement loops dominate the cost structure (87.7\% of total costs) primarily because only the Code Agent utilizes the more expensive `claude-3-5-sonnet-20241022` model for its superior coding capabilities, while all other system components employ the more cost-effective `claude-3-5-haiku-20241022` model. This design choice optimizes both performance and cost efficiency by allocating premium computational resources specifically to the most demanding implementation tasks.
>
> The time cost of 4.796 hours encompasses both LLM inference response time and agent experimental execution time across all system components. These measurements validate the practical feasibility of our approach while demonstrating the computational investment required for high-quality iterative code refinement in autonomous scientific research.
>
> ## Test-Time Scaling Capabilities
>
> | **Variants**         | **Correctness** |
> | -------------------- | --------------- |
> | AI-Researcher iter 3 | 3.2221          |
> | AI-Researcher iter 2 | 2.8888          |
> | AI-Researcher iter 1 | 2.4443          |
>
> We apologize for overlooking this important question about our test-time scaling claim. Thank you for pointing out the lack of empirical support for this statement.
>
> To address this concern, we conducted additional experiments on test-time scaling using our AI-Researcher system with Claude-series models on Level-1 tasks in the vector quantization domain. We varied the number of Code-Advisor iterative loops as our scaling metric.
>
> As shown in Table 1, our results demonstrate clear test-time scaling capabilities: performance increases progressively with more iterative refinement cycles, from 2.44 (1 iteration) to 2.89 (2 iterations) to 3.22 (3 iterations). This empirical evidence supports our claim that additional computational investment at test-time through iterative refinement cycles leads to measurably improved implementation quality.
>
> This scaling behavior validates our architectural design where each additional Code-Advisor loop provides structured feedback and refinement opportunities, enabling the system to progressively correct implementation errors and improve code quality. The consistent performance gains across iterations demonstrate that our framework can effectively leverage additional test-time computation for enhanced research outcomes.
>
> We will incorporate these test-time scaling results into the main paper to provide proper empirical support for our claims.

---

> ### Author Response · Authors · 2025-08-02
> **Further Response to Reviewer XKkH (2/2)**
>
> ## Distinguishing Execution Failures from Scientific Quality Issues
>
> Thank you for the thoughtful follow-up. We appreciate your interest in how the system handles cases where the output, while executable, may not meet scientific or methodological expectations.
>
> In our dual-metric evaluation framework (Section 4.1), we make a clear distinction between **execution feasibility** and **implementation quality**. Specifically, **“Unfeasible”** refers to cases where the agent fails to produce runnable code after multiple refinement cycles—commonly due to persistent issues such as tensor shape mismatches, NaN losses, or out-of-memory (OOM) errors.
>
> In contrast, cases where the implementation completes successfully but exhibits **scientific flaws, conceptual inconsistencies, or inadequate hypothesis validation** fall under the **Correctness** dimension. These are not marked as “Unfeasible” because the task has technically been completed. Instead, the Advisor Agent analyzes such outcomes and flags potential weaknesses in reasoning or methodology, and the Judge Agent reflects these issues in a lower correctness score on a 5-point scale.
>
> In summary, while "Unfeasible" denotes hard execution failure, deeper quality issues—including but not limited to “failed hypotheses”—are systematically captured by our **Correctness** evaluation process.

---

> > ### Comment · Reviewer_XKkH · 2025-08-05
> > **response**
> >
> > I appreciate the authors' detailed responses, which address my initial concerns. As a result, I am raising my score, assuming these changes and clarifications will be incorporated into the revised paper.

---

### Decision · Program_Chairs · 2025-09-17

**Decision:**

Accept (spotlight)

**Comment:**

The paper presents a multi-agent system designed to automate the full scientific research pipeline—from literature review and hypothesis generation to algorithm implementation and manuscript writing—largely without human intervention. To evaluate such systems, the authors introduce Scientist-Bench, a benchmark based on 22 state-of-the-art AI research papers spanning guided and open-ended tasks. Experiments demonstrate that AI-Researcher achieves high implementation success rates and produces outputs approaching human-level quality, with especially strong performance on open-ended research tasks. The paper positions itself as both a framework for autonomous research and a benchmark for future developments in this domain.

The work is timely and ambitious, addressing the important problem of automating scientific discovery. Reviewers praised the end-to-end design, the systematic evaluation framework, and the strong empirical results, particularly in open-ended tasks. Weaknesses centered on clarity, overstated claims of producing “publication-ready” manuscripts, reliance on LLM-as-a-judge without sufficient human validation, missing comparisons to related frameworks, and limited discussion of ethical risks. In their rebuttal, the authors addressed many of these concerns by clarifying unclear mechanisms, providing ablations and comparisons to other frameworks, introducing a pilot human expert evaluation that confirmed alignment with LLM-based judgments, and reporting cost and runtime analysis. Post-rebuttal, reviewers acknowledged that these additions substantially strengthened the paper, though questions remain regarding the generality of the system beyond AI research and the robustness of LLM-based evaluation under adversarial conditions.

Given its novelty, the thorough improvements made in the rebuttal, and the consensus among reviewers that it is technically solid with high potential impact, this paper should be accepted.